# Multi-cohort study identifies social determinants of systemic inflammation over the life course

Eloïse Berger[1], Raphaële Castagné[1], Marc Chadeau-Hyam[2], Murielle Bochud[3], Angelo d'Errico[4], Martina Gandini[4], Maryam Karimi[2], Mika Kivimäki[5,6], Vittorio Krogh[7], Michael Marmot[5], Salvatore Panico[8], Martin Preisig[3], Fulvio Ricceri[4], Carlotta Sacerdote[9], Andrew Steptoe[5], Silvia Stringhini[10], Rosario Tumino[11], Paolo Vineis[2,12], Cyrille Delpierre[1] & Michelle Kelly-Irving[1]

Chronic inflammation has been proposed as having a prominent role in the construction of social inequalities in health. Disentangling the effects of early life and adulthood social disadvantage on inflammation is key in elucidating biological mechanisms underlying socioeconomic disparities. Here we explore the relationship between socioeconomic position (SEP) across the life course and inflammation (as measured by CRP levels) in up to 23,008 participants from six European cohort studies from three countries conducted between 1958 and 2013. We find a consistent inverse association between SEP and CRP across cohorts, where participants with a less advantaged SEP have higher levels of inflammation. Educational attainment is most strongly related to inflammation, after adjusting for health behaviours, body mass index and later-in-life SEP. These findings suggest socioeconomic disadvantage in young adulthood is independently associated with later life inflammation calling for further studies of the pathways operating through educational processes.

[1] LEASP, UMR 1027, Inserm-Université Toulouse III Paul Sabatier, Toulouse 31000, France. [2] MRC-PHE Centre for Environment and Health, School of Public Health, Department of Epidemiology and Biostatistics, Imperial College London, London W2 1PG, UK. [3] Psychiatric Epidemiology and Psychopathology Center, Department of Psychiatry, Lausanne University Hospital, Lausanne 1004, Switzerland. [4] Epidemiology Unit, ASL TO3 Piedmont Region, Grugliasco 10095, Italy. [5] Department of Epidemiology and Public Health, University College London, London WC1E 6BT, UK. [6] Clinicum, Faculty of Medicine, University of Helsinki, P. O. Box 20 Helsinki FI-00014, Finland. [7] Epidemiology and Prevention Unit, Fondazione IRCCS Istituto Nazionale dei Tumori, Milan 20133, Italy. [8] Department of Clinical Medicine and Surgery, University of Naples Federico II, Naples 80131, Italy. [9] Piedmont Reference Centre for Epidemiology and Cancer Prevention (CPO Piemonte), Turin 10126, Italy. [10] Institute of Social and Preventive Medicine, Lausanne University Hospital, Lausanne 1010, Switzerland. [11] Cancer Registry and Histopathology Department, 'Civic – M. P. Arezzo' Hospital, ASP Ragusa, Ragusa 97100, Italy. [12] Molecular and Genetic Epidemiology Unit, Italian Institute for Genomic Medicine (IIGM), Torino 10126, Italy. These authors contributed equally: Eloïse Berger, Raphaële Castagné. These authors jointly supervised: Paolo Vineis, Cyrille Delpierre, Michelle Kelly-Irving. Correspondence and requests for materials should be addressed to M. K.-I. (email: michelle.kelly@inserm.fr)

Heightened systemic inflammation has been linked to many chronic diseases[1]. Inflammation is a set of responses that may be caused by a number of processes such as the presence of an infection, central-adipose tissue, or tumour cell development, but also occur as a consequence of the chronic solicitation of the stress response system, namely neuroinflammation or sterile inflammation[2,3]. A higher level of basal inflammation has consequences for overall health and has been linked to mortality across various causes[4,5]. CRP is an acute-phase protein synthesized by the liver in response to systemic effects of inflammation[6] and is generally considered a marker of overall inflammatory response. CRP has been considered as a marker of atherosclerosis and coronary heart disease, playing an important role in the formation of plaques in arterial walls and triggering of a cardiac or cerebrovascular event[7]. Prospective epidemiologic studies suggest that elevated circulating levels of CRP are also associated with an increased risk of certain types (lung, colorectal), but not all cancers[8].

Underlying the aetiological drivers of inflammation implicated in ageing and chronic diseases is the backdrop of the socio-economic environment. A number of studies highlight socio-economic disadvantage as an upstream determinant of increased basal inflammation. A systematic review of 25 population-based studies reported that low SEP mainly assessed by education was associated with elevated CRP level in adulthood across countries[9]. Elevated levels of others circulating inflammatory markers were also reported in disadvantaged socioeconomic groups in general[10–13] and also regarding gender differences[14]. A recent meta-analysis of 15 studies focusing on SEP in childhood revealed an inverse association between early life SEP through parental education or occupation and adulthood CRP[15]. These relationships exemplified the biological embodiment, or the social-to-biological transition[16] which could be an important set of processes and mechanisms involved in the construction of health inequalities over the life course. Several processes can explain the impact of the social environment on inflammation. First, socially disadvantaged populations are disproportionately exposed to environments that can be characterized as pro-inflammatory[17]. This includes exposure to infections due to overcrowded conditions, poor housing quality, or insufficient access to sanitation[18]. Second, adverse health behaviours, more prevalent among disadvantaged socioeconomic groups, may expose individuals to pro-inflammatory factors, such as tobacco smoking as well as nutritional behaviours leading towards central obesity[19,20]. Finally, cumulative social disadvantage may also lead individuals towards experiencing adversities, or exacerbate such situations, and result in an over-solicited stress response system, which in turn, contribute to heightened basal inflammation[21,22].

Social exposures occur from childhood and across life course stages consisting of both biological and socially sensitive periods. They may lead to early wear-and-tear of physiological systems and ultimately to worsened health[23–25]. In the available literature on the relationship between SEP and inflammation, the influence played by country and period-specific contexts on the social patterning of inflammatory response has been given limited attention due to the lack of available data and/or cross-country variable harmonisation. Investigating the temporal nature of social exposures over the life course and the inflammatory response later in life also needs to be better elucidated. Unpicking these questions may highlight mechanisms through which the socioeconomic gradient in health emerges over the life course, how to prevent it from emerging, and how to mitigate the effects of disadvantage on inflammatory processes underlying morbidity.

This study aims to examine several important aspects of the social-to-biological transition using harmonised data across six European cohort studies collected within the Lifepath consortium (Supplementary Note 1)[26]. Our approach consists of taking into account the chronology of exposures over the life course, to understand how they are associated with inflammation. First, we assessed the relationship between life course SEP (father's occupation, educational attainment, participant's last occupation) at three time points individually and systemic inflammation using the biomarker C reactive protein (CRP) measured in adulthood. We then investigated the potential impact of behavioural factors and body mass index (BMI) on this relationship. Second we investigated life course effects of SEP experiences by sequentially controlling for time-ordered SEP. To gain a better understanding of the gender and cohort effects, analyses were performed separately within each cohort for men and women. Cohort-specific estimates were combined using a random effect meta-analysis.

## Results

**Study population.** The sample selection for each cohort is provided in Fig. 1 along with a description of the main participants characteristics by cohort in Table 1. Key characteristics of each study population showed small to moderate differences compared to our analytical sample (Supplementary Data 1). Complete data on CRP, two life course SEP variables (educational attainment and last occupation) as well as intermediate factors were available for 23,008 [55.7% of men] participants across the 6 cohorts and 13,078 [59.6% of men] participants when restricting to the 4 cohorts where father's occupation was available). Mean age ranged between 45.8 (SD 1) years of age in NCDS and 67.4 (SD 9.4) in ELSA. Between 46.8% (NCDS) and 51.7% (ELSA) of participants were women except for Whitehall II (29.2%). Mean serum CRP levels (SD) ranged from 1.9 (4.3) mg/L in Whitehall II to 4.0 (7.8) in ELSA. The proportion of participants with a low educational attainment ranged from 38.8% in Whitehall II to 76.3% in NCDS. The largest proportion of participants that reported drinking more than 21 (men) or 14 (women) alcoholic units per week was observed in NCDS (26.8%); additionally, EPIC-Italy had the largest proportion of smokers (29.6%), 28.1% of the participants from ELSA were obese and 40.6% of Skipogh participants were sedentary.

Descriptive characteristics for each study by SEP are shown in Supplementary Data 2. Variation was observed in participant's distribution in terms of their characteristics, by life course SEP and by study. Despite this, in general participants with a less advantaged SEP were more likely to smoke (except in EPIC-Italy) and systematically more likely to be overweight/obese and sedentary (except for EPIC-Italy) compared to the most advantaged. Alcohol consumption followed a heterogeneous pattern both by study and SEP.

**Association between each respective SEP indicator and CRP.** Results for childhood SEP (measured by father's occupational position) are given in Table 2A and Fig. 2a. Despite the moderate to high degree of heterogeneity in study-specific estimates, results from meta-analysis revealed a significant association between less advantaged childhood SEP and higher CRP level in adulthood in the overall population (Model 1: Less advantaged vs More advantaged $\beta = 0.19$, $P < 0.001$) Controlling for either alcohol consumption, smoking or sedentary lifestyle had little effect on the observed association, whereas adjustment for BMI attenuated associations between SEP and CRP (Model 1 + BMI: Less advantaged vs More advantaged $\beta = 0.10$, $P = 0.005$) leading to a coefficient attenuation of 47.4% (Supplementary Note 2). Controlling for all intermediate factors in adulthood explained part but not all of the observed association between childhood SEP and CRP level in adulthood which remained significant (Model 2: Less advantaged vs More advantaged $\beta = 0.08$, $P = 0.021$).

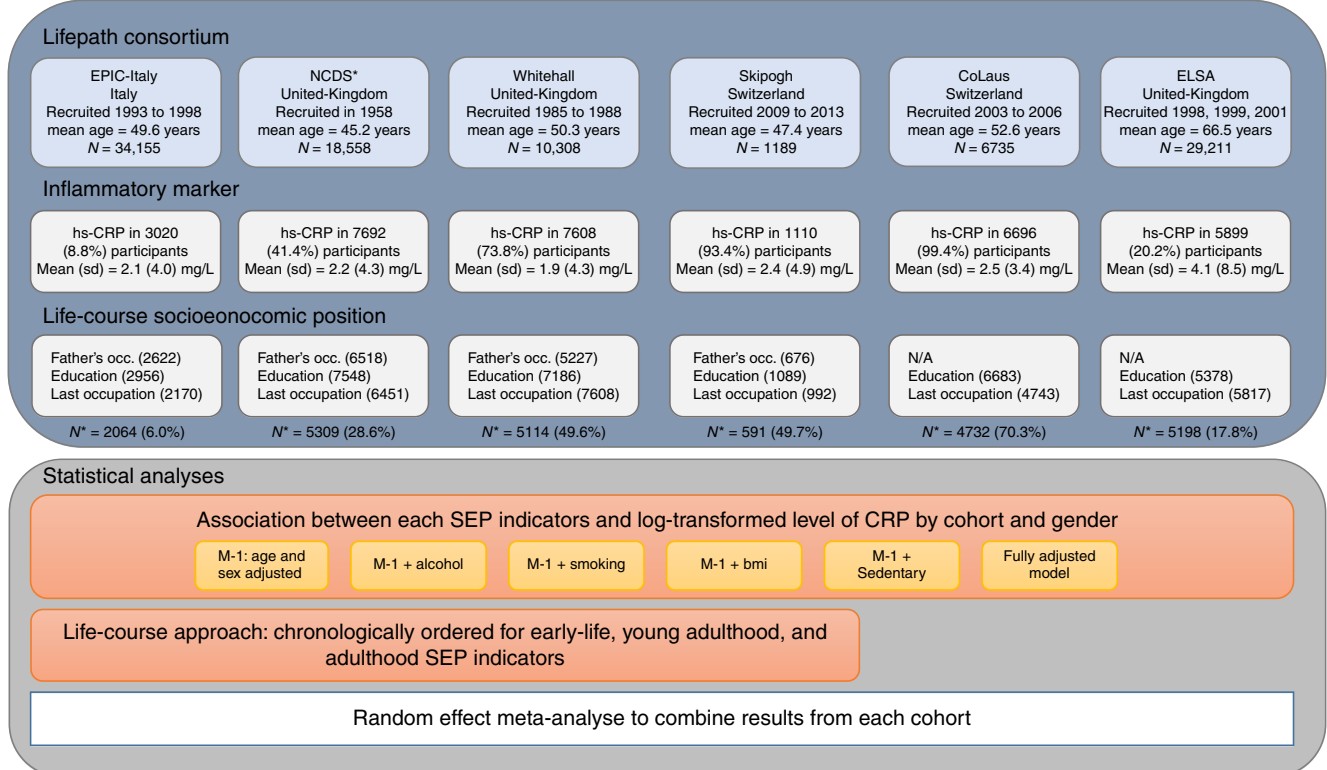

**Fig. 1** Overview of the study workflow. *NCDS is the only birth cohort, therefore father's occupation was collected prospectively and mean age corresponds to the age of participants at the time of the biomedical survey

A strong, graded association was observed between educational attainment and CRP (Table 2B and Fig. 2b). Results from the random effects meta-analyses revealed significant associations between low educational level and higher CRP level in the overall population (Model 1: Low vs High $\beta = 0.30$, $P < 0.001$). Observed associations remained markedly stable upon adjustment for alcohol consumption, smoking and sedentary lifestyle, but were weakened after adjustment for BMI (Model 1 + BMI: Low vs High: $\beta = 0.19$, $P < 0.001$) corresponding to a regression coefficient attenuation of 36.7% (Supplementary Note 2). The association between educational attainment and CRP, though attenuated, remained significant after adjustment for the four intermediates factors (Model 2: Low vs High $\beta = 0.14$, $P < 0.001$). In general, the strength of the associations between a medium level of education and CRP were weaker but significant compared to those with a low level of education, with a gradient persisting across educational groups in relation with CRP.

Having a less advantaged occupational position in adulthood was associated with having a higher CRP consistently in the random effect meta-analyses in the overall sample (Table 2C and Fig. 2c, Model 1 Less advantaged vs More advantaged: $\beta = 0.24$, $P < 0.001$). Among the four potential intermediate factors tested, the relationship between adulthood SEP and CRP was mostly affected by BMI in the random effect meta-analysis overall (Model 1 + BMI $\beta = 0.17$, $P < 0.001$) leading to a coefficient attenuation of 29.2% (Supplementary Note 2). The observed relationship between adulthood SEP and CRP was weakened when fully-adjusted (Model 2: $\beta = 0.11$, $P < 0.001$).

**Contribution of SEP at each stage of the life course to adult CRP.** We sequentially controlled for time-ordered life course SEP in the subset of four cohorts with complete data for SEP indicators over the life course (father's occupation, education and occupation during adulthood), i.e. Skipogh, EPIC-Italy, Whitehall

and NCDS. This allowed us to evaluate the contribution of each specific SEP variable to examine their relative impact by life course stage (childhood, young adulthood and older adulthood), and by type of SEP measure (educational attainment or occupation). The association between less advantaged early-life SEP and CRP in adulthood (Model A $\beta = 0.19$, $P < 0.001$) was weakened when educational attainment (Model B $\beta = 0.11$, $P = 0.006$) or later occupation (Model C $\beta = 0.16$, $P = 0.001$) or both (Model D $\beta = 0.10$, $P = 0.011$) were included in the model and no longer significant in the fully adjusted model (Table 3).

An independent relationship between low educational attainment and high CRP was observed regardless of the model (Model B $\beta = 0.24$, $P < 0.001$, fully adjusted model $\beta = 0.12$, $P = 0.048$).

Participant's last occupation was no longer significantly associated with CRP when controlling for education (Model D $\beta = 0.04$, $P = 0.693$), the association being almost null after adjustment for the four intermediate factors (Fully adjusted model $\beta = 0$, $P = 0.998$).

**Gender effect.** A significant association between less advantaged childhood SEP and higher CRP was observed in men (Model 1: Less advantaged vs More advantaged $\beta = 0.12$, $P = 0.011$, Supplementary Table 1A) and women (Model 1: Less advantaged vs More advantaged $\beta = 0.28$, $P < 0.001$, Supplementary Table 2A). The observed relationship between adulthood SEP and CRP was still significant albeit weakened after controlling for the four potential intermediate factors tested in women only (Model 2: Less advantaged vs More advantaged $\beta = 0.12$, $P = 0.046$, Supplementary Table 2A).

The strong, graded association was observed between educational attainment and CRP in men (Model 1: Low vs High $\beta = 0.28$, $P < 0.001$, Supplementary Table 1B) and women (Model 1: Low vs High $\beta = 0.33$, $P < 0.001$, Supplementary Table 2B). The association between educational attainment and CRP, though

| Table 1 Descriptive statistics of the participants for the six cohort studies ($N = 23,008$ participants) | | | | | | | |
|---|---|---|---|---|---|---|---|
| | Skipogh ($N = 591$) | EPIC-Italy ($N = 2064$) | CoLaus ($N = 4732$) | Whitehall ($N = 5114$) | ELSA ($N = 5198$) | NCDS ($N = 5309$) | All cohorts ($N = 23,008$) |
| | Mean (SD) or N (%) | Mean (SD) or N (%) | Mean (SD) or N (%) | Mean (SD) or N (%) | Mean (SD) or N (%) | Mean (SD) or N (%) | Mean (SD) or N (%) |
| *Demographics* | | | | | | | |
| Age | 48.8 (15.7) | 52.6 (8.1) | 49.2 (8.8) | 49.9 (6) | 67.4 (9.4) | 45.8 (1) | 53 (11) |
| Sex—Men | 286 (48.39) | 1071 (51.89) | 2502 (52.87) | 3618 (70.75) | 2509 (48.27) | 2823 (53.17) | 12,809 (55.67) |
| Women | 305 (51.61) | 993 (48.11) | 2230 (47.13) | 1496 (29.25) | 2689 (51.73) | 2486 (46.83) | 10,199 (44.33) |
| *Inflammation* | | | | | | | |
| CRP (mg/L) | 2.6 (4.8) | 2 (2.7) | 2.2 (3.2) | 1.9 (4.3) | 4 (7.8) | 2 (4) | 2.5 (5) |
| *Socioeconomic position* | | | | | | | |
| *Father's occupation* | | | | | | | |
| Most advantaged | 130 (22) | 110 (5.33) | NA | 487 (9.52) | NA | 325 (6.12) | 1052 (8.04) |
| Middle | 212 (35.87) | 838 (40.6) | NA | 1561 (30.52) | NA | 873 (16.44) | 3484 (26.64) |
| Less advantaged | 249 (42.13) | 1116 (54.07) | NA | 3066 (59.95) | NA | 4111 (77.43) | 8542 (65.32) |
| *Educational attainment* | | | | | | | |
| High | 113 (19.12) | 234 (11.34) | 1665 (35.19) | 1740 (34.02) | 732 (14.08) | 984 (18.53) | 5468 (23.77) |
| Medium | 188 (31.81) | 476 (23.06) | 668 (14.12) | 1387 (27.12) | 1130 (21.74) | 273 (5.14) | 4122 (17.92) |
| Low | 290 (49.07) | 1354 (65.6) | 2399 (50.7) | 1987 (38.85) | 3336 (64.18) | 4052 (76.32) | 13,418 (58.32) |
| *Last occupation* | | | | | | | |
| Most advantaged | 110 (18.61) | 131 (6.35) | 798 (16.86) | 2619 (51.21) | 1861 (35.8) | 1752 (33) | 7271 (31.6) |
| Middle | 201 (34.01) | 1012 (49.03) | 1495 (31.59) | 1626 (31.8) | 1804 (34.71) | 1690 (31.83) | 7828 (34.02) |
| Less advantaged | 280 (47.38) | 921 (44.62) | 2439 (51.54) | 869 (16.99) | 1533 (29.49) | 1867 (35.17) | 7909 (34.38) |
| *Intermediate factors* | | | | | | | |
| *Alcohol consumption* | | | | | | | |
| Abstainer | 171 (28.93) | 265 (12.84) | 1215 (25.68) | 964 (18.85) | 2314 (44.52) | 1043 (19.65) | 5972 (25.96) |
| High | 113 (19.12) | 352 (17.05) | 405 (8.56) | 805 (15.74) | 1303 (25.07) | 1427 (26.88) | 4405 (19.15) |
| Low | 307 (51.95) | 1447 (70.11) | 3112 (65.77) | 3345 (65.41) | 1581 (30.42) | 2839 (53.48) | 12,631 (54.9) |
| *Smoking status* | | | | | | | |
| Current | 149 (25.21) | 611 (29.6) | 1366 (28.87) | 723 (14.14) | 736 (14.16) | 1272 (23.96) | 4857 (21.11) |
| Former | 209 (35.36) | 619 (29.99) | 1509 (31.89) | 1909 (37.33) | 2397 (46.11) | 1473 (27.75) | 8116 (35.27) |
| Never | 233 (39.42) | 834 (40.41) | 1857 (39.24) | 2482 (48.53) | 2065 (39.73) | 2564 (48.3) | 10,035 (43.62) |
| *BMI* | | | | | | | |
| Underweight | 12 (2.03) | 6 (0.29) | 46 (0.97) | 29 (0.57) | 28 (0.54) | 16 (0.3) | 137 (0.6) |
| Normal weight | 294 (49.75) | 763 (36.97) | 2371 (50.11) | 2648 (51.78) | 1424 (27.4) | 2557 (48.16) | 10,057 (43.71) |
| Overweight | 201 (34.01) | 959 (46.46) | 1702 (35.97) | 1950 (38.13) | 2287 (44) | 2030 (38.24) | 9129 (39.68) |
| Obese | 84 (14.21) | 336 (16.28) | 613 (12.95) | 487 (9.52) | 1459 (28.07) | 706 (13.3) | 3685 (16.02) |
| *Sedentary* | | | | | | | |
| Yes | 240 (40.61) | 440 (21.32) | 1666 (35.21) | 991 (19.38) | 1429 (27.49) | NA | 4766 (26.93) |
| No | 351 (59.39) | 1624 (78.68) | 3066 (64.79) | 4123 (80.62) | 3769 (72.51) | NA | 12,933 (73.07) |

attenuated, remained significant upon adjustment on the four intermediates factors in men (Model 2, Supplementary Table 1B) and women (Model 2, Supplementary Table 2B).

Men and women with a less advantaged SEP in adulthood had higher CRP levels compared to their more advantaged counterparts when analysed separately (Model 1 in men: Less advantaged vs more advantaged $\beta = 0.21$, $P = 0.001$, Model 1 in women: Less advantaged vs More advantaged $\beta = 0.31$, $P < 0.001$, Supplementary Table 1C and Table 2C, respectively). Regarding the life course SEP analyses, a similar pattern was observed in men as well as women (Supplementary Tables 3 and 4).

**Heterogeneity between cohorts**. Regarding cohort-specific findings, depicted in Supplementary Fig. 1, we observed that participants whose father had a middle or less advantaged occupational position had a significantly higher level of CRP in Skipogh, EPIC-Italy, Whitehall, and NCDS, but no longer significant after adjustment for all intermediate factors except in EPIC-Italy (Supplementary Table 5).

The association between low educational attainment and CRP was consistent across cohorts (Supplementary Fig. 2 and Supplementary Table 6). This association was slightly attenuated upon adjustment for BMI but remained significant in CoLaus ($\beta = 0.14$, $p < 0.001$), Whitehall ($\beta = 0.11$, $p = 0.002$), ELSA ($\beta = 0.33$, $p < 0.001$), NCDS ($\beta = 0.28$, $p < 0.001$) and for CoLaus, ELSA and NCDS in the fully adjusted model ($\beta = 0.10$, $p = 0.003$; $\beta = 0.23$, $p < 0.001$; $\beta = 0.22$, $p < 0.001$, respectively, Supplementary Table 6).

Participants with less advantaged occupational position had a significantly higher level of CRP compared to those with a more advantaged occupation in CoLaus, Whitehall, ELSA and NCDS (Supplementary Fig. 3 and Supplementary Table 7). Among the four potential intermediate factors tested, the relationship between adulthood SEP and CRP was mostly affected by BMI remaining significant in CoLaus, Whitehall, ELSA, NCDS. The fully adjusted model followed the same pattern albeit weakened and the association in NCDS was no longer significant.

Regarding the life course SEP analysis, educational attainment was significantly associated with CRP in each cohort separately (Model B, Supplementary Table 8) but the adjustment for subsequent mediators played a different role according cohort: education was no longer significant in the fully adjusted model in Whitehall but still significant in NCDS.

**Table 2 Multiple regression analyses of (A) father's occupational position in 4 cohorts (B) participant's educational attainment and (C) participant's last occupation with CRP at baseline in six cohorts from the Lifepath project. Meta-analyses results for the total population includes N = 13,078 for early life SEP and N = 23,008 for later in life SEP, except for Model 1 + sedentary where N = 7,769 and N = 17,699 respectively**

| | Category | (A) Father's occupational position[a] | | (B) Participant's educational attainment[b] | | (C) Participant's last occupation[a] | |
| --- | --- | --- | --- | --- | --- | --- | --- |
| | | Middle (26.6%) | Less advantaged (65.3%) | Medium (17.9%) | Low (58.3%) | Middle (34.0%) | Less advantaged (34.4%) |
| Model 1[c] | $\beta$ (95% CI) | 0.11 (−0.07; 0.28) | 0.19 (0.11; 0.27) | 0.15 (0.08; 0.21) | 0.30 (0.22; 0.38) | 0.09 (0.01; 0.16) | 0.24 (0.14; 0.35) |
| | P-value | 0.235 | <0.001 | <0.001 | <0.001 | 0.023 | <0.001 |
| | $I^2$ | 74.8% | 1.3% | 43.4% | 75.6% | 67.9% | 81.4% |
| | $P_H$ | 0.015 | 0.466 | 0.147 | 0.001 | 0.009 | <0.001 |
| Model 1[c] + Alcohol | $\beta$ (95% CI) | 0.10 (−0.08; 0.28) | 0.18 (0.11; 0.26) | 0.14 (0.08; 0.20) | 0.29 (0.21; 0.36) | 0.08 (0.004; 0.15) | 0.22 (0.12; 0.33) |
| | P-value | 0.260 | <0.001 | <0.001 | <0.001 | 0.038 | <0.001 |
| | $I^2$ | 76.2% | 0.0% | 32.7% | 68.5% | 67.0% | 81.0% |
| | $P_H$ | 0.012 | 0.518 | 0.262 | 0.008 | 0.010 | <0.001 |
| Model 1[c] + Smoking | $\beta$ (95% CI) | 0.11 (−0.07; 0.28) | 0.18 (0.10; 0.25) | 0.13 (0.07; 0.20) | 0.27 (0.19; 0.35) | 0.07 (0.01; 0.14) | 0.21 (0.12; 0.31) |
| | P-value | 0.233 | <0.001 | <0.001 | <0.001 | 0.036 | <0.001 |
| | $I^2$ | 75.2% | 0.0% | 41.8% | 74.6% | 62.3% | 79.1% |
| | $P_H$ | 0.014 | 0.452 | 0.167 | 0.001 | 0.026 | <0.001 |
| Model 1[c] + BMI | $\beta$ (95% CI) | 0.06 (−0.10; 0.22) | 0.10 (0.03; 0.17) | 0.10 (0.03; 0.16) | 0.19 (0.11; 0.28) | 0.08 (0.02; 0.14) | 0.17 (0.09; 0.25) |
| | P-value | 0.493 | 0.005 | 0.003 | <0.001 | 0.014 | <0.001 |
| | $I^2$ | 73.4% | 0.0% | 41.1% | 79.3% | 55.7% | 73.3% |
| | $P_H$ | 0.019 | 0.451 | 0.171 | <0.001 | 0.047 | 0.004 |
| Model 1[c] + Sedentary | $\beta$ (95% CI) | 0.16 (−0.002; 0.32) | 0.17 (0.08; 0.26) | 0.13 (0.06; 0.19) | 0.25 (0.18; 0.32) | 0.12 (0.07; 0.16) | 0.26 (0.21; 0.32) |
| | P-value | 0.053 | 0.000 | <0.001 | <0.001 | <0.001 | <0.001 |
| | $I^2$ | 54.2% | 0.0% | 31.7% | 60.4% | 0.2% | 18.5% |
| | $P_H$ | 0.109 | 0.381 | 0.296 | 0.048 | 0.488 | 0.099 |
| Model 2[d] | $\beta$ (95% CI) | 0.05 (−0.11; 0.21) | 0.08 (0.01; 0.15) | 0.07 (0.02; 0.13) | 0.14 (0.07; 0.21) | 0.05 (0; 0.10) | 0.11 (0.05; 0.17) |
| | P-value | 0.518 | 0.021 | 0.006 | <0.001 | 0.051 | <0.001 |
| | $I^2$ | 74.1% | 0.0% | 21.4% | 67.2% | 35.4% | 51.4% |
| | $P_H$ | 0.017 | 0.458 | 0.411 | 0.009 | 0.176 | 0.065 |

CI confidence interval $I^2$ percentage of between study heterogeneity, $P_H$ P-value of heterogeneity test, BMI body mass index
[a] Referent group: most advantaged
[b] Referent group: high educational attainment
[c] Model 1 adjusted for age and sex
[d] Model 2 controlled for age, sex and additionally alcohol, smoking, BMI and sedentary

**Sensitivity analyses**. The exclusion of participants with a CRP ≥10 mg/L (N = 870, 3.8%) had little effect on the previous observed associations between life course SEP and CRP: a social gradient in adult CRP was still observed. Regarding educational attainment, participants with low educational level had a significant higher level of CRP (Model 1 meta-analysis $\beta = 0.26$, $P < 0.001$, Supplementary Table 9). Our results were slightly attenuated but still significant after adjustment for BMI (Model 1 + BMI meta-analysis $\beta = 0.17$, $P < 0.001$, Supplementary Table 9) and in the fully adjusted model (fully adjusted model meta-analysis $\beta = 0.12$, $P < 0.001$, Supplementary Table 9).

In Skipogh, CoLaus, Whitehall and ELSA, CRP measures were also collected after the first follow-up (Supplementary note 3): the strong, graded association between educational level and CRP was still observed few years later (fully adjusted meta-analysis: $\beta = 0.10$, $P = 0.012$ and $\beta = 0.15$, $P = 0.001$ for mid and low educational level, respectively, Supplementary Table 10).

**Discussion**
In this multi-cohort study using data from several European countries, we found that disadvantaged socioeconomic position at each life stage was associated with increased inflammation assessed using CRP. As a general pattern, accounting for behavioural factors (alcohol consumption, smoking status and sedentary) and BMI explained part, but not all of the observed SEP differences in inflammation. In general, associations between life course SEP and inflammation were mainly attenuated by BMI in adulthood. A subsequent analysis carried out whereby life course SEP variables were added to the model in chronological order indicated that participants with a low educational attainment had a higher level of CRP independently of early life or later adulthood SEP as well as behavioural factors and BMI. The persistence of a significant association between low educational attainment and a high CRP level in adulthood indicates that educational attainment may be a consistent and important upstream risk factor for elevated inflammation. These findings suggest that in addition to health behaviours and BMI, alternative pathways by which SEP may affect inflammation deserve to be studied further. These include, for example, chemical exposures, infectious diseases (such as Herpes and Epstein Barr infections etc) and oral health conditions, as well as psychosocial stress.

Overall, our results are consistent with previous studies investigating the relationship between SEP at different life stages and chronic inflammation. A previous systematic review of published observational studies up to 2006 reported associations between adult SEP and CRP[9]. Since then the same relationship has also been demonstrated in various other observational studies from worldwide countries (Taiwan[27], Europe[13], Brazil[28] and others[17,29–31]). A recent meta-analysis of population-based and cross-sectional studies showed that low childhood SEP was associated with higher adulthood CRP[15].

Though we observed significant associations in some of the fully adjusted models, notably for disadvantaged educational

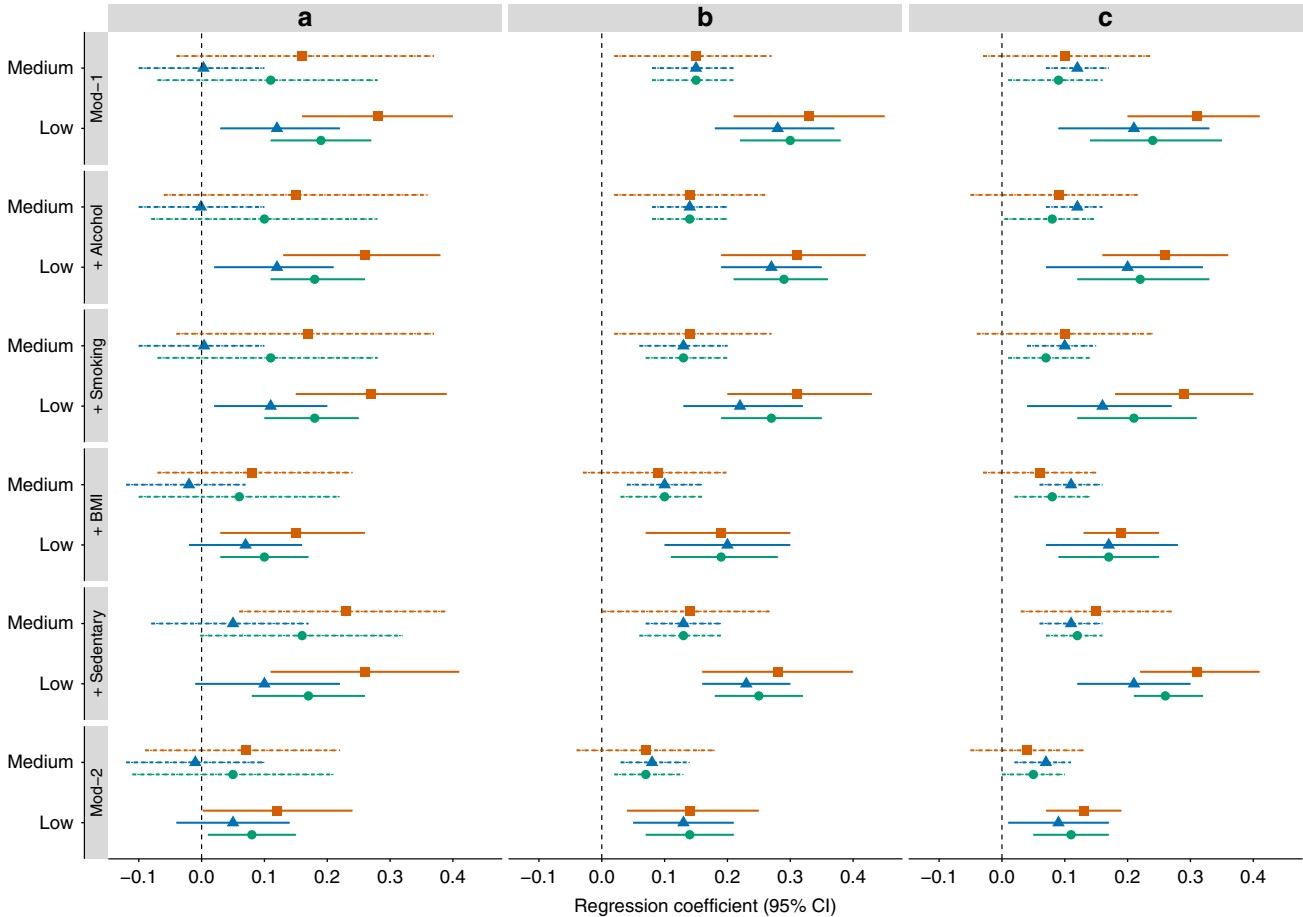

**Fig. 2** Forest plot of regression coefficients [95% confidence interval] for the association between **a** father's occupational position, **b** participant's educational attainment, **c** participant's last occupation and CRP concentration at baseline in random effect meta-analysis framework for the total population and by gender for Model 1 (Mod-1), after adjustment for each intermediate factor (+Alcohol, +Smoking, +BMI, +Sedentary) and further adjusted for all intermediate factors together (Mod-2). The high SEP group was used as reference, solid lines represent the medium SEP group and dotted lines the low SEP group. Meta-analyses results for the total population (in orange) includes $N = 13,078$ for early life SEP and $N = 23,008$ for later in life SEP, respectively, $N = 7798$ and $N = 12,809$ for men (in blue) and $N = 5280$ and $N = 10,199$ for women (in green)

attainment and SEP, adjusting for intermediate factors weakened associations. This may reflect indirect pathways between SEP and inflammation. The intermediate factor that most influenced the association between SEP and CRP was BMI. Other studies have reported that BMI attenuates the relationships between SEP and CRP[28,32]. Together these sources of evidence may point towards the accumulation of body fat among more socially disadvantaged populations as being a mechanism leading to higher levels inflammation. Nutrient excess is related to dysregulation in the cellular and molecular intermediate factors of immunity and inflammation. With an estimated excess of 20–30 million macrophages accumulate with each kilogram of excess fat in humans, one could argue that increased adipose tissue mass is a state of increased inflammatory mass[33]. Smoking is also a known inflammation-inducing behaviour[34]. In the meta-analyses, alterations in the association between SEP and CRP when smoking was introduced were negligible. In the overall meta-analyses alcohol consumption did not affect the results once added to the models, nor did sedentary lifestyle. This phenomenon could be explained by the fact that smoking and alcohol do not always show a clear social gradient contrary to BMI which is consistently associated with SEP across all cohorts. As expected, the results indicate that the social distribution of intermediate factors is key to their ability to act as a mediators between SEP and health.

Beyond behaviours and BMI, the association between less advantaged SEP and CRP remained, notably when educational attainment was examined. This may suggest other underlying social-to-biological mechanisms operating whereby the socio-economic environment confers inflammation-inducing or reducing mechanisms. One of the likely mechanisms involved is the stress response system, where higher educational attainment may act as a physiological stress-regulating buffer. In the case of education, a higher educational attainment may provide increased sense of control, which is a suggested pathway linking education to health[35]. Though gaps in the literature remain as to whether psychological measures such as sense of control relate to inflammation[36,37], such relationships have recently been described for multiple stress biomarkers (known as sterile inflammation) and for real-life stress exposure[38,39]. Emotional regulation has been associated with both academic success[40] in children and inflammation[41]. A more advantaged SEP may also favour access to better paid[42] more stable occupations with access to work flexibility.

Overall, we found similar patterns in the relationship between SEP and CRP for men and women, though the effect size between father's occupation and educational attainment and CRP were higher in women. Some differences between men and women may come down to differences in immunological and inflammatory responses influenced by both sex and gender. Sex contributes to

**Table 3 Life course multiple regression analyses of SEP with CRP at baseline in a random effect meta-analytical framework from 4 cohorts from the Lifepath (N = 13,078)**

| | Model Aᵃ | | | | | Model Bᵇ | | | | |
|---|---|---|---|---|---|---|---|---|---|---|
| | β | 95% CI | P-value | I² | P_H | β | 95% CI | P-value | I² | P_H |
| *Father's occupation* | | | | | | | | | | |
| Middle | 0.11 | (−0.07; 0.28) | 0.235 | 74.8% | 0.015 | 0.06 | (-0.11; 0.23) | 0.489 | 73.0% | 0.020 |
| Less advantaged | 0.19 | (0.11; 0.27) | <0.001 | 1.3% | 0.466 | 0.11 | (0.03; 0.19) | 0.006 | 0.0% | 0.585 |
| *Educational level* | | | | | | | | | | |
| Medium | – | – | – | – | – | 0.10 | (0.03; 0.16) | 0.003 | 0.0% | 0.640 |
| Low | – | – | – | – | – | 0.24 | (0.14; 0.35) | <0.001 | 64.5% | 0.026 |
| *Last occupation* | | | | | | | | | | |
| Middle | – | – | – | – | – | – | – | – | – | – |
| Less advantaged | – | – | – | – | – | – | – | – | – | – |
| **Model Cᶜ** | | | | | | **Model Dᵈ** | | | | |
| Father's occupation | | | | | | | | | | |
| Middle | 0.09 | (−0.09; 0.27) | 0.306 | 76.2% | 0.013 | 0.06 | (−0.12; 0.24) | 0.501 | 75.0% | 0.015 |
| Less advantaged | 0.16 | (0.07; 0.26) | 0.001 | 28.7% | 0.270 | 0.10 | (0.02; 0.18) | 0.011 | 0.0% | 0.480 |
| *Educational level* | | | | | | | | | | |
| Medium | – | – | – | – | – | 0.10 | (0.01; 0.18) | 0.021 | 19.2% | 0.433 |
| Low | – | – | – | – | – | 0.23 | (0.09; 0.38) | 0.002 | 78.9% | 0.000 |
| *Last occupation* | | | | | | | | | | |
| Middle | 0.02 | (−0.10; 0.14) | 0.758 | 73.9% | 0.002 | −0.03 | (−0.17; 0.12) | 0.706 | 79.3% | 0.000 |
| Less advantaged | 0.14 | (−0.02; 0.31) | 0.084 | 82.6% | 0.000 | 0.04 | (−0.17; 0.25) | 0.693 | 88.2% | <0.001 |
| **Fully adjustedᵉ** | | | | | | | | | | |
| *Father's occupation* | | | | | | | | | | |
| Middle | 0.04 | (−0.13; 0.21) | 0.658 | 76.0% | 0.012 | | | | | |
| Less advantaged | 0.05 | (−0.02; 0.12) | 0.192 | 0.1% | 0.359 | | | | | |
| *Educational level* | | | | | | | | | | |
| Medium | 0.05 | (−0.03; 0.12) | 0.231 | 18.3% | 0.467 | | | | | |
| Low | 0.12 | (0.001; 0.23) | 0.048 | 69.8% | 0.008 | | | | | |
| *Last occupation* | | | | | | | | | | |
| Middle | −0.02 | (−0.12; 0.08) | 0.683 | 60.9% | 0.038 | | | | | |
| Less advantaged | 0.00 | (−0.13; 0.13) | 0.998 | 70.8% | 0.012 | | | | | |

CI confidence interval, I² heterogeneity, P_H P-value of heterogeneity test, BMI body mass index
ᵃ Model A adjusted for age, sex, father's occupational position
ᵇ Model B adjusted for age, sex, father's occupational position and participant's educational attainment
ᶜ Model C adjusted for age, sex, father's occupational position and participant's last occupation
ᵈ Model D adjusted for age, sex, father's occupational position, participant's educational attainment and participant's last occupation
ᵉ Fully adjusted model controlled age, sex, father's occupational position, participant's educational attainment, participant's last occupation and additionally alcohol, smoking, BMI and sedentary

differences that influence the deposit of adipose tissue, its physiological interaction with the endocrine system, and associated metabolic risk over the life course[43]. Gender may reflect behaviours that influence exposure to microorganisms, access to healthcare or health-related behaviours that affect health trajectories[44]. We also observed that the attenuation of the relationship once BMI was introduced into the models appeared to be greater among women. This may suggest that inflammation induced through greater body mass and adipose tissue was a more significant pathway for women from less advantaged SEP groups[45].

Findings were directionally consistent across the cohort studies from the UK, Italy and Switzerland, representing a variety of contexts and period, although differences in effect size were observed. This suggests that the social-to-biological process we are observing is shared across European high-income country populations. Some differences between cohorts are likely down to their specific characteristics. The strongest association between father's occupation and CRP were observed for the NCDS and Skipogh even though the NCDS reported among the highest proportions of manual occupation among participant's fathers, while Skipogh had the lowest. Whitehall II, and occupational cohort of British civil-servants had one of the lowest proportions of respondents with a manual occupation, and the relationship between manual occupation and CRP was the strongest among the cohorts we studied. BMI has also a dominant role across cohorts and for all SEP variables (Supplementary Figs. 1–3; Supplementary Tables 5-7) even though the prevalence of obese individuals by SEP differed by cohort study: ELSA, a study of ageing in an elderly English population, was the cohort with the lowest prevalence of normal BMI, and with the highest proportion of obese individuals compared to the other cohorts. These results highlight the important to take into account the social distribution of risk factors in a specific population to understand which of them may contribute to the relationship between SEP and biology.

The main strength of the study was the use of harmonised data from six cohort studies on participants from three European countries. The longitudinal nature of the studies allowed us to examine how SEP at different stages of the life course preceding the measurement of CRP are related. This allows us to begin unpacking the complex relationships between social and biological variables by measuring them at different time points. Given the varied nature of the cohorts and their populations, we were able to maintain the assumption that the relationships being assessed in the models were cohort specific by using a random effects model when pooling the cohorts together. The breadth and depth of data available allowed us to perform multiple sensitivity analyses, which underlined further the stability of the results. Finally, the use of CRP as a reliable and reproducible biomarker measuring overall inflammatory response allows for comparisons, given its wide availability as a biomarker through high-sensitivity assays which are stable.

All cohort studies are susceptible to attrition, and biased samples. The cohorts we used may be not representative of the general population, in particular because of a potential under-representation of the more advantaged/less advantaged SEP categories. In these analyses, we used complete case data from each dataset to conduct cohort-specific and pooled meta-analyses using random effects modelling. Our models included a set of harmonised variables available across the cohorts allowing us to adjust for covariates, however there may be some residual confounding. This possibility of confounding means that we cannot fully ascertain causal directionality. Furthermore, the harmonisation process favours standardising variables across cohorts, which may mean that some cohort-specificities are smoothed out, or lost. This process also led to some variation in our models and results (only 4 cohorts contained father's occupation and a lack of equivalent harmonised variable for sedentary lifestyle for the NCDS). There is likely to be measurement error and heterogeneity across cohorts regarding socioeconomic variables. Father's occupation was collected in a variety of ways, referring to historical and

country specific occupations, and ultimately these variables were divided in three categories. This is likely to have led to some degree of misclassification. Additionally, the distribution of the SEP measures was also skewed. To assess the robustness of our results and since misclassification error may have occurred for the assessment of life course SEP, we repeated the main analysis (i) using the Less advantaged/Low group as a reference (Supplementary Table 11) and (ii) with a binary indicator of father's occupational position and participant's last occupation (Supplementary Table 12). Estimates of the associations were all consistent with those in the main analysis, suggesting that our findings are robust. The SEP indicators we used measure partially different and inter-dependent aspects of life experiences and may be related to different perceptions and belief about health-related behaviours. Education, for instance, reflects the transition from childhood/adolescence to adulthood together with the intellectual and socioeconomic resources of the family, but it is also an important determinant of future employment and income[46,47].

To meet the assumption that CRP follows a Gaussian distribution in the sample, values were transformed on the natural log scale to approximate the normal curve; alternative techniques to address the skewness of the CRP distribution include parametric modelling based on flexible size distributions[48]. We only had one inflammation biomarker that was available across all these cohorts where life course SEP variables had been collected, therefore the generalisation to others inflammatory-related biomarkers remains to be shown. While CRP is a useful proxy for overall inflammatory status, we would ideally have used it alongside Tumour Necrosis Factor α and Interleukin 6 for example, in order to better capture the complex regulatory cascades involved in inflammation. The inflammatory response is coordinated by a number of cytokines as well as signalling proteins of the immune system[49]. Future investigations on inflammatory-related biomarkers at wider and larger scale are needed to disentangle the complex inflammatory responses associated to pathological and immunological processes.

Our findings provide consistent evidence across cohorts that less advantaged socioeconomic conditions experienced at three different life stages are associated with increased overall inflammation in adulthood. The findings point towards educational attainment as the SEP variable most strongly associated with CRP, suggesting that further examination of social-to-biological pathways operating through educational processes are worthy of biomedical and public health attention. Questions remain as to whether educational attainment captures material and/or psychosocial exposures affecting biology through the maturation of the immune system, having lasting effects on overall inflammation.

In conclusion, our study highlights the important role that, from early life, social factors play in health beyond behaviours and lifestyle factors. We document the biological consequences – through inflammation – of social disadvantages, justifying the need to intervene from early life to reduce social disparities in health. Further work integrating biosocial perspectives in longitudinal settings is needed to better understand the mechanisms through which the social environment shapes physiological processes that are important for complex health outcomes.

## Methods

**Study population**. Lifepath is a Horizon 2020 Research and Innovation Program European funded project which aims to investigate the biological pathways underlying social differences in healthy ageing. For this purpose, the project includes a consortium of eighteen cohort studies (child and adult) across different countries and time periods, with demographic, clinical, biological and socioeconomic data. Data harmonization were performed in order to merge and analyse the cohorts together[50]. More details about the Lifepath project are available elsewhere (http://www.lifepathproject.eu/).

We selected six cohorts for which data on CRP, at least two measures of socioeconomic position at two different life course stages, and behaviours/BMI were available. Detailed information on cohorts used in our study is provided in Supplementary Table 13. Briefly, our panel of cohorts includes a subset of the Italian component of the European prospective investigation on nutrition and cancer study (EPIC-Italy)[51]; two cohorts from Switzerland: CoLaus[52] and Skipogh[53]; and three British cohorts: Whitehall II[54], the English longitudinal study of ageing (ELSA)[55], and the National Child Development Study (NCDS)[56]. All Lifepath cohorts have been described in detail elsewhere[50]. Each study was approved by the relevant local or national ethics committees and all participants gave informed consent to participate.

**Inflammatory markers**. We used CRP to measure overall inflammation, selecting the first available measurement of CRP in each cohort for our analyses. Information about sampling methods and laboratory analyses for each cohort study are given in Supplementary Table 15. Baseline CRP was measured in mg/L using high-sensitivity assay in all studies. CRP values were natural log transformed in order to normalise their distributions given their skewed nature.

**Life course socioeconomic positions**. We used the harmonised definition of life course SEP across cohort study described in details by d'Errico et al.[26]. Childhood SEP was ascertained using father's occupational position reported by the participants and recoded according to the European Socioeconomic Classification (E-SeC) where occupations are classified according to their employment relations and work conditions. We applied 3 E-SeC categories: less advantaged occupations [lower clerical, services, and sales workers, skilled workers, semi-skilled and unskilled workers, E-SeC class 7–9], middle occupations [small employers and self-employed, farmers, lower supervisors and technicians, E-SeC class 4–6], and more advantaged occupations [higher professionals and managers, higher clerical, services, and sales workers, E-SeC class 1–3].

SEP in young adulthood was measured using the participant's educational attainment categorised in three groups: primary or lower secondary school (from 7 to 9 years after kindergarten with a basic curriculum in languages, mathematics and other subjects); higher secondary school (around 4–5 years more, high school diploma level) and tertiary education (any degree after high school, such as BSc, MSc, and further education)[26] hereafter referred as low, medium and high educational level respectively. Adulthood SEP was measured by the participant's last occupational position in three groups: less advantaged occupations, middle occupations, and more advantaged occupations according to the same three occupational classes used for father's occupational position and based on self-reported information.

**Health behaviours and lifestyle factors**. The following factors have been found to be associated with both SEP and CRP in the literature and were therefore considered as intermediate variables that may mediate the relationship between SEP and CRP: BMI[57,58], categorical: <18.5 kg/m$^2$, underweight; 18.5–24.9 kg/m$^2$, normal; 25.0–29.9 kg/m$^2$, overweight; >18.5 kg/m$^2$, obese); smoking status[34,59] (categorical: current, former, never); alcohol consumption[60,61] (categorical: abstainers, moderate consumption – men: ≤21 alcohol units per week/women: ≤14 alcohol units per week but not abstainers, high consumption – men: >21 alcoholic units per week/women: >14 alcoholic units per week); sedentary[62,63] (categorical: sedentary, no sedentary based on response to questions on leisure physical activity) except in NCDS in which this variable was not collected. We selected these intermediate variables from the closest data collection wave to that of the CRP measurement. If data were unavailable at the same wave, we imputed data from the next available data wave (for CoLaus: $n = 12$, 0.3%; Skipogh: $n = 159$, 26.9%; Whitehall: $n = 277$, 5.4%; ELSA: $n = 69$, 1.3%).

**Statistical analysis**. Means and frequencies were reported for all continuous and categorical baseline characteristics by cohort and by each SEP variable (father's occupation, educational attainment, participant's last occupation). Chi-squared test or Fisher exact test for the categorical variables and T-test or Wilcoxon rank test for continuous variables were used to estimate bivariate associations.

Linear regression models were used to investigate the relationship between SEP and CRP concentration at baseline. We defined a minimally adjusted model controlling for age and sex (sex only for the NCDS where participants are the same age) (Model 1), then we further adjusted for each potential intermediate factor independently. We finally defined a second model including all potential intermediate factors (Model 2). For all three SEP indicators, the highest socioeconomic group was used as reference (non-manual for father's job and last occupation and high for educational level). A positive regression coefficient therefore indicates an increased level of CRP in less advantaged socioeconomic groups.

To mimic life course experiences, we sequentially adjusted for the chronologically ordered SEP indicators; resulting in four time-sequenced models on the subset of the four cohorts (Skipogh, EPIC-Italy, Whitehall and NCDS) containing all three SEP indicators:

Model A: Father's occupation + Age + Sex
Model B: Father's occupation + Educational attainment + Age + Sex

Model C: Father's occupation + Last occupation + Age + Sex

Model D: Father's occupation + Educational attainment + Last occupation + Age + Sex

Fully adjusted: Model D + Intermediate factors

Model A allows us to identify the potential early life biological embedding of social conditions, while model B and C allows the identification of emerging social-to-biological signals specific to young adult (model B) and later adulthood (model C) experiences. The resulting regression coefficients measure mutually adjusted effects. Model D evaluates the relative contribution of each SEP measure across the life course. The fully adjusted model estimates the effect of each SEP indicator, controlled for the intermediate factors in adulthood.

None of the interaction tested between gender and SEP was found significant (Supplementary Table 14) within each cohort studies. But prior research on health and health behaviours have indicated that there may be important gender differences pattern by SEP[64]. Evidence are also accumulating showing potential sex differences along the life course process of disease development and progression. Furthermore physiological responses to chronic stress could differently impact men and women. So far few studies have examined gender in relation to SEP and CRP in cohort studies setting[14,65,66]. Therefore, the multivariate linear analysis was run separately by gender.

Random effects meta-analyses[67] were conducted using the metafor R package. Between-study heterogeneity was estimated through a restricted maximum-likelihood estimator. The combined effect represents the mean of the population of true effects. We reported the estimated average effect ($\hat{\mu}$) and the estimated percentage of the total amount of variability that can be attributed to heterogeneity ($I^2$) together with the Q-statistic associated $p$-value.

We repeated the analyses excluding individuals with CRP above 10 mg/L, as high CRP maybe more likely result from an acute infection than chronic inflammation (Supplementary Table 9). Analyses were repeated using CRP measured at the next follow-up when available (Skipogh, CoLaus, Whitehall and ELSA, Supplementary Table 10). We ran all the analyses using the less advantaged/low group as a reference for each life course SEP (Supplementary Table 11) and using dichotomized SEP indicators for father's occupational position and participant's last occupation (Supplementary Table 12)

Statistical analyses were performed using R. version 1.1.383.

**Reporting summary**. Further information on experimental design is available in the Nature Research Reporting Summary linked to this article.

## Data availability

The data and computer code uses to support the findings of this study are available from the corresponding author upon reasonable request.

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

## Acknowledgements

This study was supported by the European Commission Horizon 2020 grant number 633666 to Paolo Vineis. Thank you to the Lifepath Consortium (see Supplementary Note 1). We are also grateful to The Centre for Longitudinal Studies, UCL Institute of Education for the use of these data and to the UK Data Archive and UK Data Service for making them available. M.K. is supported by the MRC (K013351), the Academy of Finland (311492) and NordForsk. The Centre for Environment and Health is supported by the Medical Research Council and Public Health England (MR/L01341X/1).

## Author contributions

C.D., P.V., and M.K.-I. designed research; M.B., A.E., M.G., M.Ka., M.Ki., V.K., M.M., S.P., M.P., F.R., C.S., A.S., S.S., R.T., P.V. collected the data; M.B., A.E., M.G., M.Ka., M.Ki., V.K., M.M., S.P., M.P., F.R., C.S., A.S., S.S., R.T., P.V. contributed reagents, materials or analysis tools; E.B., R.C., M.C.-H., C.D. and M.K.-I. analysed the data; E.B., R.C., P.V., C.D., and M.K.-I. discussed the results and wrote the paper. All authors reviewed and commented on the manuscript.

## Additional information

**Competing interests:** The authors declare no competing interests.

