## [Peer Review File · Nature Communications]

Reviewers' Comments:

Reviewer #1:

Remarks to the Author:

Social determinants of systemic inflammation over the life course: a multi-cohort study

This is an original, very interesting study on the association between socio-economic position and inflammation across the life course. The study employs robust statistical analysis, and the detailed sensitivity analysis is an additional advantage for the study.

I have the following minor comments and suggestions:

- 1) Abstract: explicitly define the SEP measures as well as name the countries and the time horizon that the study covers.
- 2) Graphs at the main text: improve the quality of the graphs (particularly the relevant text; for example, p-values etc.).
- 3) Subtitles "Social effect of life course SEP on CRP" and "Life course SEP and CRP": more clarity is needed (for example, by modifying/editing the subtitles) in order to further highlight the distinct analyses/inferences that are relevant to each part of the paper.
- 4) Table 2: there is some need for a better visualisation of panels A, B and C.
- 5) Table 3: explain/clarify the difference between model D and the "fully adjusted" model.
- 6) Although the study addresses the skewness of the CRP distribution by using log-transformed models (which is excellent), the authors may consider citing recent research on alternative estimation techniques that may be used for highly skewed biomarker distributions:

Davillas, A., & Jones, A. M. (2018). Parametric models for biomarkers based on flexible size distributions. *Health economics*.

Reviewer #2:

Remarks to the Author:

Social determinants of systemic inflammation over the life course: a multi-cohort study

This is an interesting report on the analysis of several well-known cohorts regarding the association of several socioeconomic (SE) indicators with C-reactive protein (CRP). This can provide evidence on the biological substrate of socioeconomic health differences. Next, I have three main concerns, and in addition I think that the Discussion could be better developed.

First, in their analyses, they use a very small category of favourable as reference category, leading to three questions: how do the authors categorise a given SE position as favourable etc.? How can the favourable category be so small in these cohorts? And why do they take such a small category as reference (making estimates really rather unstable)?

Second, they infer that their outcome, CRP, represents a systemic inflammation being the pathway to cardiovascular, cancer, and other diseases. These all regard rather life-threatening diseases but why would the elevated CRP in the cohorts as assessed not simply be caused by chronic inflammation of the mouth, in particular the gingiva? SE differences in oral health are very well-known, and evidence is strong that this leads to an increased level of CRP. The authors should at least discuss and preferably be able to exclude this explanation. Otherwise, they are simply confirming one of the best known diseases attributable to the determinants that they assess. They may refer to this in their

remark in the discussion that 'These findings also suggest that other pathways by which SEP may affect CRP levels may exist beyond health behaviours and BMI.' (line 220 and further). Please substantiate.

Third, the authors use the label of life course but do not really substantiate that the three indicators which they use only indicate one stage of the life course. This at least is not the case regarding a person's education: that does not only indicate his or her early adult SE position, but may be inferred as indicating the SE position of that person in later life as well. Some discussion of this issue should be included.

More in detail, I have a number of additional issues, and some specification of the above.

Introduction

The authors do not cite the recent meta-analysis of Liu et al. (J Epidemiol Community Health. 2017;71:817-826) on childhood SES and CRP levels in adulthood which in fact summarises evidence on a much larger number of participants (n=43 thousand), albeit not with the primary data.

The justification of adjusting for some health behaviours should be strengthened. Why would e.g. sedentary behaviour lead to an increased CRP?

Results

Figure 2 presents results of three analyses (A) to (C) – it would be nice to repeat the labels of these in the figure itself.

Next, as indicated at the start, the percentages of unfavourable are very high, e.g. >60% for father's occupational status, and also for educational level (the same applies to Table 1). This seems to be rather incredible, the analyses now take a very small category as reference. The authors do not specify how they classified the various categories. This should be specified, and such a large category 'unfavourable' seems very unlikely to me. Looking to the Supplementary tables, this issue becomes even larger. E.g. for father's occupational status, in EPIC Italy, only 5.3% of the population is advantaged and in NCDS 6.1% (Supplementary Figure 1).

In Table 3, the authors should preferably present findings for each of the three SE indicators separately -they currently do not so for Educational level. Further, the Fully adjusted model should be explained in a footnote – what does the adjustment entail ?

Discussion

The authors are right in entering the issue of contextual influences given the sometimes pretty large between-country differences. Next, the long paragraph concerned (lines 223-244) mostly repeats findings that have already been presented in the Results without adding interpretation, except some words on the Whitehall II study. This requires a redoing.

A similar criticism applies to the paragraph on smoking (lines 259-267): many repeats of findings, but what do we learn from this?

The paragraph on educational attainment and emotion regulation (lines 268-279) is highly speculative, too speculative in its current format. E.g. a sentence like 'Though gaps in the literature remain as to whether psychological measures such as sense of control relate to 276 inflammation, such relationships have been described for stress hormones, such as cortisol.' without any referencing should not be in.

Materials & Methods

The authors present many detailed analyses which I appreciate. Next, more detail is needed on the handling on the variables, regarding the way in which the independent variables (i.e. determinants) have been categorised. Simply denoting occupations as "disadvantaged occupations", "middle occupations", and "advantaged occupations" is not reproducible and should be substantiated.

Regarding education, the authors should definitely use an international classification like the international standard classification of education (ISCED).

Regarding the outcome, CRP: did the laboratories concerned use similar laboratory standards, i.e. how occurred the calibration?

References

Regarding references, some of them are really very old without being classics, e.g. ref. 18 (1997) and ref. 40 (1974). Ref. 19 and 22 seem to misspell the name of Soc Sci Med (adding 1982 to it) and ref. 24 is from 2016 but still in press. Ref. 37 misses information on its publication source.

Reviewer #3:

Remarks to the Author:

Overall comments

The authors of this paper show a consistent inverse association between low socioeconomic position (SEP) across life and higher levels of CRP in 4-6 cohorts (3 from the UK, 2 from Switzerland and 1 from Italy) with up to 23,000 participants. There is considerable interest in this field both from scientists interested in how social factors get 'under the skin' and those interested in common life course processes underlying ageing. The associations revealed are not novel but this multi-cohort study strengthens the evidence base. There has been a previous systematic review of SEP and CRP in 2007 which the authors only cite in the discussion (Nazmi and Victoria, their ref 28) which looked at over 30 studies (including 4 with early life SEP), reached similar conclusions, and recommended more studies of early SEP and adult inflammation. I would suggest the authors should cite this study in the introduction and explain how their research builds on this systematic review. Their acknowledgement of the previous literature in the discussion is also somewhat limited and undermines the novel aspects that their study brings. For example, I am aware that life course SEP and adult inflammatory markers (both CRP and IL-6) and the role of BMI have been investigated in the 1946 British birth cohort (Jones R, et al. Novel Coronary Heart Disease Risk Factors at 60-64 years and Life Course Socioeconomic Position. *Atherosclerosis*. 2015 Jan;238(1):70-6) with similar results, yet this paper (and maybe others) are not cited, and I suggest they should be. Similarly, in the discussion, acknowledging that gender differences in lifetime SEP exposure on adult BMI have been previously published would help them explain the gender differences in the associations between SEP and inflammation, and the stronger attenuation by BMI in women.

In regards to the statistical analysis, if, as the authors indicate, they are taking a life course approach to investigate the association between lifetime SEP and adult inflammation, could the authors explain why they do not use the structured life course modelling approach (which has been used by several groups) to select the life course models that best fit the data, rather than a simple set of regression models? Also gender differences are reported and commented upon but no tests of interaction are mentioned. A statistical reviewer may have other points but this is not my area of expertise.

The results, figures and tables could do with clarification in places and better labelling (see below). The discussion is wordy and could be tightened considerably, and repetitions removed. Previous studies that support (or not) the findings of this study should be adequately cited.

So overall this is a useful study because of the multi-cohort features but it somewhat undermines itself by not showing how it builds on previous literature. It confirms previous evidence rather than influencing new thinking in this field. The conclusions need to be strengthened to be more precise about the overall strength of the evidence and the further research needed and then what the clinical

or policy implications could/should be.

Specific points

Abstract: The abstract would be strengthened by: including a brief rationale for studying the association between lifetime SEP and later CRP; including the key estimates from the regression models; making the last sentence less vague. (see also typo 'adpoting')

Introduction:

p.4-5, lines 92-93: Suggest the phrase 'the social patterning of the life course response remains poorly described in terms of life course stage' needs toning down (given overall comments above) and relevant references should be cited here and then commented upon in the discussion.

Last paragraph: the introduction would be strengthened by: stating some hypotheses based on what we already know about this topic; and by making clear that when the analyses were performed separately for men and women, tests of sex differences would also be conducted.

Results

First para of text. Figure 1 needs improving. What the mean age refers to should be clear in the figure – it appears from the text to be age at enrolment but that is not correct for NCDS at least. Include the standard deviation as well as the mean CRP level. How do those included in the samples (ranging from 6%-70%) differ from those who were not included in the samples? Perhaps also make clear which measures of father's occupation were prospective and retrospective. ELSA CRP levels are almost twice as high compared with others – is this the result of their higher mean age?

Line 128 subheading 'Social effect of life course SEP on CRP' is a rather curious heading. And it is not clear how it differs from the second subheading 'life course SEP and CRP'. Rephrase?

Line 129-160 I found these three paragraphs hard to follow with the relevant tables. The title of the tables need better labelling (so it is clear they are showing the various adjusted models) and better signposting in the text. When the text indicates that the associations are stronger in men than women (or vice versa) have tests of interaction been performed – perhaps give relevant statistic in brackets?

Discussion

216-217 'to mimic life course experiences' needs clarifying given that the materials and methods come later.

Line 230-231 – reference to previous literature not sufficient (see overall comments). Whole paragraph is wordy, needs tightening.

Line 245-258 – finding that BMI attenuates the associations most is not a novel finding and other studies should be cited to provide the strength of the evidence overall.

Line 289-291 There are a number of papers that need citing that show that the accumulation of SEP exposures across life on adult overweight/obesity are stronger for women than men which may explain the gender differences and also why the attenuation by BMI was stronger in women than men.

Line 303-307: so how did those with missing data in the cohorts differ from those with complete covariate data?

319-321 A definite limitation is the use of CRP as the only inflammatory marker. This limitation needs more discussion (e.g. in relation to the inflammatory cascade) and whether other inflammatory markers (like IL-6) could be more informative.

Line 325-329: so were the SEP indicators associated with BMI in each of the cohorts?

Line 330-337: the last paragraph of the discussion raises rather vague needs for further research. Can the authors be more precise and also comment on any clinical or policy implications. Some overall assessment of the strength of the evidence (not just their study) would be helpful.

Materials and methods

359-360: Information about sampling methods and laboratory analyses are not given in Supplementary table 8.

377-379: the authors say that the choice of intermediate variables was based on previous literature but fail to cite these studies.

Statistical analysis

Nowhere does it indicate that sex interaction tests were carried out. Please clarify as statements that some associations were stronger in men than women (or vice versa) are made in the text.

Line 402-416: Please explain whether the structured life course modelling approach would have been preferable to models A-C to test whether lifetime SEP exposures were cumulative or linked to sensitive periods.

Reviewer #1 (Remarks to the Author):

Social determinants of systemic inflammation over the life course: a multi-cohort study

This is an original, very interesting study on the association between socio-economic position and inflammation across the life course. The study employs robust statistical analysis, and the detailed sensitivity analysis is an additional advantage for the study.

We thank the reviewer for this positive feedback on our work.

I have the following minor comments and suggestions:

1) Abstract: explicitly define the SEP measures as well as name the countries and the time horizon that the study covers.

As suggested by Reviewer 1 we clarified the abstract and provided details about SEP, countries and the time horizon covered by the different cohort studies.

2) Graphs at the main text: improve the quality of the graphs (particularly the relevant text; for example, p-values etc.).

In the revised manuscript, as suggested by Reviewer 1, we have improved the quality of the graphs.

3) Subtitles “Social effect of life course SEP on CRP” and “Life course SEP and CRP”: more clarity is need (for example, by modifying/editing the subtitles) in order to further highlight the distinct analyses/inferences that are relevant to each part the paper.

We agree with the Reviewer 1 that our results subsections headings were rather confusing and have corrected them accordingly “Association between each respective SEP indicator and CRP” and “Contribution of SEP at each stage of the life course to adult CRP”

4) Table 2: there is some need for a better visualisation of panels A, B and C.

Table 2 in the revised version of the manuscript has been improved to ease the distinction between panels A, B, and C. Table 2 in the revised version of the manuscript has been transposed to give in columns each SEP indicators and in rows corresponding results for each model. A detailed footnote has also been added to better describe the table

[†]Except for model 1 + sedentary where N = 7 769

[‡]Except for model 1 + sedentary where N = 17 699

^{*}Model 1 adjusted for age and sex

[‡]Model 2 controlled for age, sex and additionally alcohol, smoking, BMI and sedentary

Abbreviations: CI, confidence interval; I^2 , percentage of between study heterogeneity; P_H , P-value of heterogeneity test; BMI, body mass index. “

5) Table 3: explain/clarify the difference between model D and the “fully adjusted” model.

The Table 3 in the revised version of the manuscript has been clarified. A detailed footnote has been added to better describe the linear regression models

***Model A adjusted for age, sex, father's occupational position*

[†]Model B adjusted for age, sex, father's occupational position and participant's educational attainment

[‡]Model C adjusted for age, sex, father's occupational position and participant's last occupation

[§]Model D adjusted for age, sex, father's occupational position, participant's educational attainment and participant's last occupation

[¶]Fully adjusted model controlled age, sex, father's occupational position, participant's educational attainment, participant's last occupation and additionally alcohol, smoking, BMI and sedentary

Abbreviations: CI, confidence interval; I², heterogeneity; PH, P-value of heterogeneity test; BMI, body mass index. "

6) Although the study addresses the skewness of the CRP distribution by using log-transformed models (which is excellent), the authors may consider citing recent research on alternative estimation techniques that may be used for highly skewed biomarker distributions:

Davillas, A., & Jones, A. M. (2018). Parametric models for biomarkers based on flexible size distributions. *Health economics*.

As suggested, we have added the following sentence in the Discussion:

"To meet the assumption that CRP follows a Gaussian distribution in the sample, values were transformed on the natural log scale to approximate the normal curve; alternative techniques to address the skewness of the CRP distribution include parametric modelling based on flexible size distributions ⁴⁶."

46: Davillas, A. & Jones, A. M. Parametric models for biomarkers based on flexible size distributions. Health Econ. (2018). doi:10.1002/hec.3787

Reviewer #2 (Remarks to the Author):

Social determinants of systemic inflammation over the life course: a multi-cohort study

This is an interesting report on the analysis of several well-known cohorts regarding the association of several socioeconomic (SE) indicators with C-reactive protein (CRP). This can provide evidence on the biological substrate of socioeconomic health differences.

We thank the reviewer for this positive comment.

Next, I have three main concerns, and in addition I think that the Discussion could be better developed.

First, in their analyses, they use a very small category of favourable as reference category, leading to three questions:

R.1. How do the authors categorise a given SE position as favourable etc.?

This has now been clarified in the life course SEP method section (see p 17) where we provided the following description:

"Childhood SEP was ascertained using father's occupational position reported by the participants and recoded according to the European Socioeconomic Classification (E-SeC) where occupations are

classified according to their employment relations and work conditions. We applied 3 E-SeC categories: “disadvantaged occupations” [lower clerical, services, and sales workers, skilled workers, semi-skilled and unskilled workers, E-SeC class 7-9], “middle occupations” [small employers and self-employed, farmers, lower supervisors and technicians, E-SeC class 4-6], and “advantaged occupations” [higher professionals and managers, higher clerical, services, and sales workers, E-SeC class 1-3].”

In addition, we have added details to the entire section.

R.2. How can the favourable category be so small in these cohorts?

Life course SEP characteristics of our study population show small to moderate differences from the originating population for each cohort study (Supplementary Table 1), suggesting that the impact of missingness does not explain the small proportion of the advantaged category in these cohorts.

The small proportion of advantaged participants (<10%) was mostly observed for father occupational position in EPIC-Italy, NCDS and Whitehall cohort studies, which are also those who recruited their participants at the late nineties (up to 1998) suggesting a possible cohort effect. In these cohorts, the question relating to the father's occupation refers to a period after the Second World War when the proportion of socially advantaged people, as defined in our study (see above), was low. Additionally, potential bias arising from the data harmonisation cannot be excluded.

R.3. And why do they take such a small category as reference (making estimates really rather unstable)?

Our reference category was sufficiently large for analyses with robust results. More specifically, in order to test our primary hypothesis predicting that low SEP is inversely associated with CRP level in adulthood, the advantaged/high category was used as a reference for father's/last occupational position and educational attainment respectively in multivariate regression models. Results from the random effect meta-analysis revealed a significant association between (a) disadvantaged childhood SEP and higher CRP level in adulthood in the overall population (Model 1: Disadvantaged vs Advantaged $\beta=0.19$, $P<0.001$; Middle vs Advantaged $\beta=0.11$, $P=0.235$) (b) low educational level and higher CRP level in the overall population (Model 1 Low vs High $\beta=0.30$, $P<0.001$; Middle vs Low $\beta=0.15$, $P<0.001$) and (c) disadvantaged adulthood SEP and higher CRP level in the overall population (Model 1: Disadvantaged vs Advantaged $\beta=0.24$, $P<0.001$; Middle vs Advantaged $\beta=0.09$, $P=0.023$).

To address the reviewer's point, we additionally repeated the main analyses using the disadvantaged/low group as the reference (Supplementary Table 13). Regression coefficients of the association between Advantaged (High) vs Disadvantaged (Low) were logically similar but with reversed signs. For both participant's occupational attainment and last occupation, significant negative associations were observed between Middle vs Disadvantaged (respectively Model 1 Middle vs Disadvantaged $\beta=-0.14$, $P<0.001$ and $\beta=-0.16$, $P<0.001$).

Since misclassification error may have occurred for occupational position within each cohort study we further re-ran all analysis using an alternative binary version of father's occupational position variable and participant's last occupation variable. Again, results were only marginally changed (Supplementary Table 14).

We have added the following sentence in the Discussion section:

“To assess the robustness of our results and since misclassification error may have occurred for the assessment of life course SEP, we repeated the main analysis (i) using the Disadvantaged/Low group

as a reference (Supplementary Table 13) and (ii) with a binary indicator of father's occupational position and participant's last occupation (Supplementary Table 14). Estimates of the associations were all consistent with those in the main analysis, suggesting that our findings are robust.” (p 15)

Second, they infer that their outcome, CRP, represents a systemic inflammation being the pathway to cardiovascular, cancer, and other diseases. These all regard rather life-threatening diseases but why would the elevated CRP in the cohorts as assessed not simply be caused by chronic inflammation of the mouth, in particular the gingiva? SE differences in oral health are very well-known, and evidence is strong that this leads to an increased levels of CRP. The authors should at least discuss and preferably be able to exclude this explanation. Otherwise, they are simply confirming one of the best known diseases attributable to the determinants that they assess. They may refer to this in their remark in the discussion that ‘These findings also suggest that other pathways by which SEP may affect CRP levels may exist beyond health behaviours and BMI.’ (line 220 and further). Please substantiate.

In our models, we have adjusted for behaviours (smoking, alcohol, physical activity) and BMI, that are known to be important risk factors associated with both inflammation and SEP. Due to the lack of data, we cannot adjust for other potential variables associated with SEP and likely to influence inflammation. As pointed out by the reviewer, the inflammation in low SEP group can be explained by several processes including infection in the mouth but also viral infections that may occur during early life (like Epstein Barr or herpes viruses) or through psychosocial stress as we have previously shown [Barboza Solís et al, Adverse childhood experiences and physiological wear-and-tear in midlife: Findings from the 1958 British birth cohort. Proc Natl Acad Sci U S A. 2015 Feb 17;112(7):E738-46. doi: 10.1073/pnas.1417325112.] However, a large body of evidence has also demonstrated that low SEP is associated with higher risk of cardiovascular, cancer, and other chronic diseases associated with elevated CRP levels [Stringhini et al, Socioeconomic status and the 25 × 25 risk factors as determinants of premature mortality: a multicohort study and meta-analysis of 1.7 million men and women. Lancet. 2017 Mar 25;389(10075):1229-1237; Mackenbach et al, Trends in health inequalities in 27 European countries. Proc Natl Acad Sci U S A. 2018 Jun 19;115(25):6440-6445; Mackenbach et al, Trends in inequalities in premature mortality: a study of 3.2 million deaths in 13 European countries. J Epidemiol Community Health. 2015 Mar;69(3):207-17; discussion 205-6].

We have modified the sentence accordingly:

“These findings suggest that in addition to health behaviours and BMI, alternative pathways by which SEP may affect inflammation deserve to be studied further. These include, for example, chemical exposures, infectious diseases and conditions (such as Herpes and Epstein Barr infections, oral health), as well as psychosocial stress.” (first paragraph of the discussion, p 11)

Third, the authors use the label of life course but do not really substantiate that the three indicators which they use only indicate one stage of the life course. This at least is not the case regarding a person's education: that does not only indicate his or her early adult SE position, but may be inferred as indicating the SE position of that person in later life as well. Some discussion of this issue should be included.

We agree with the reviewer and have acknowledged this point in the revised discussion – paragraph on the Limitations (p15) noting that each SEP indicator used in our study illustrates a social process across the life course and as such do not reflect a specific/narrow window of time.

“The SEP indicators we used measure partially different and inter-dependent aspects of life experiences and may be related to different perceptions and belief about health-related behaviours. Education, for instance, reflects the transition from childhood/adolescence to adulthood together with the intellectual and socioeconomic resources of the family, but it is also an important determinant of future employment and income^{44, 45.}”

44. Galobardes, B., Shaw, M., Lawlor, D. A., Lynch, J. W. & Davey Smith, G. Indicators of socioeconomic position (part 1). *J. Epidemiol. Community Health* 60, 7–12 (2006).

45. Galobardes, B., Shaw, M., Lawlor, D. A., Lynch, J. W. & Davey Smith, G. Indicators of socioeconomic position (part 2). *J. Epidemiol. Community Health* 60, 95–101 (2006).

More in detail, I have a number of additional issues, and some specification of the above.

Introduction

The authors do not cite the recent meta-analysis of Liu et al. (*J Epidemiol Community Health*. 2017;71:817-826) on childhood SES and CRP levels in adulthood which in fact summarises evidence on a much larger number of participants (n=43 thousand), albeit not with the primary data.

Thank you. This has now been corrected. We have added the following sentence in the introduction (p 4)

“A recent meta-analysis of 15 studies focusing on SEP in childhood revealed an inverse association between early life SEP through parental education or occupation and adulthood CRP^{15.}”

15. Liu, R. S. et al. Socioeconomic status in childhood and C reactive protein in adulthood: a systematic review and meta-analysis. *J. Epidemiol. Community Health* 71, 817–826 (2017).

The justification of adjusting for some health behaviours should be strengthened. Why would e.g. sedentary behaviour lead to an increased CRP?

We welcome this excellent suggestion by the reviewer. We have added several references in the revised version of the manuscript in the Methods sections (p18) to better strengthens the justification of the intermediate factors used in our statistical analyses.

“The following factors have been found to be associated with both SEP and CRP in the literature and were therefore considered as intermediate variables that may mediate the relationship between SEP and CRP: BMI^{55,56} [...]; smoking status^{34,57} [...]; alcohol consumption^{58,59} [...]; sedentary^{60,61} [...].”

34. Shiels, M. S. et al. Cigarette Smoking and Variations in Systemic Immune and Inflammation Markers. *JNCI J. Natl. Cancer Inst.* 106, dju294–dju294 (2014).

55. McLaren, L. Socioeconomic status and obesity. *Epidemiol. Rev.* 29, 29–48 (2007).

56. Choi, J., Joseph, L. & Pilote, L. Obesity and C-reactive protein in various populations: a systematic review and meta-analysis. *Obes. Rev. Off. J. Int. Assoc. Study Obes.* 14, 232–244 (2013).

57. Giskes, K. et al. Trends in smoking behaviour between 1985 and 2000 in nine European countries by education. *J. Epidemiol. Community Health* 59, 395–401 (2005).

58. González-Reimers, E., Santolaria-Fernández, F., Martín-González, M. C., Fernández-Rodríguez, C. M. & Quintero-Platt, G. Alcoholism: a systemic proinflammatory condition. *World J. Gastroenterol.* 20, 14660–14671 (2014).

59. Mackenbach, J. P. et al. *Inequalities in Alcohol-Related Mortality in 17 European Countries: A Retrospective Analysis of Mortality Registers*. *PLoS Med.* 12, e1001909 (2015).
60. Hamer, M. et al. *Physical activity and inflammatory markers over 10 years: follow-up in men and women from the Whitehall II cohort study*. *Circulation* 126, 928–933 (2012).
61. O'Donoghue, G. et al. *Socio-economic determinants of physical activity across the life course: A 'DEterminants of Diet and Physical ACTivity' (DEDIPAC) umbrella literature review*. *PLoS One* 13, e0190737 (2018).

Results

Figure 2 presents results of three analyses (A) to (C) – it would be nice to repeat the labels of these in the figure itself.

This has now been corrected.

Next, as indicated at the start, the percentages of unfavourable are very high, e.g. >60% for father's occupational status, and also for educational level (the same applies to Table 1). This seems to be rather incredible, the analyses now take a very small category as reference. The authors do not specify how they classified the various categories. This should be specified, and such a large category 'unfavourable' seems very unlikely to me. Looking to the Supplementary tables, this issue becomes even larger. E.g. for father's occupational status, in EPIC Italy, only 5.3% of the population is advantaged and in NCDS 6.1% (Supplementary Figure 1).

We agree with the reviewer and have amended the text to carefully describe how the harmonised SEP categories were established. The "Life course socioeconomic positions" method section has been amended (p17). Please also see R.1-3 above.

In Table 3, the authors should preferably present findings for each of the three SE indicators separately -they currently do not so for Educational level. Further, the Fully adjusted model should be explained in a footnote – what does the adjustment entail ?

Table 2 and 3 have been clarified in the revised version of the manuscript. As suggested by the reviewer, Table 2 reports the multiple regression analyses from the random effect meta-analyses in the overall population of each life course SEP examined separately with a more explicit footnote.

Table 3 reports the life course multiple regression analyses of SEP with CRP from the random effect meta-analyses in the overall population sequentially adjusting for the chronologically ordered early life, young adulthood, and adulthood SEP indicators along with a detailed footnote.

Discussion

The authors are right in entering the issue of contextual influences given the sometimes pretty large between-country differences. Next, the long paragraph concerned (lines 223-244) mostly repeats findings that have already been presented in the Results without adding interpretation, except some words on the Whitehall II study. This requires a redoing.

A similar criticism applies to the paragraph on smoking (lines 259-267): many repeats of findings, but what do we learn from this?

We thank the reviewer for the helpful suggestion and have rewritten the discussion accordingly in order to avoid repetitions and insist on what our study means and add.

The paragraph on educational attainment and emotion regulation (lines 268-279) is highly speculative, too speculative in its current format. E.g. a sentence like ‘Though gaps in the literature remain as to whether psychological measures such as sense of control relate to 276 inflammation, such relationships have been described for stress hormones, such as cortisol.’ without any referencing should not be in.

We have added references and revised the sentence to provide:

“Though gaps in the literature remain as to whether psychological measures such as sense of control relate to inflammation^{36,37}, such relationships have recently been described for multiple stress biomarkers (known as ‘sterile inflammation’) and for real-life stress exposure^{38,39}.”

36. Boylan, J. M. & Ryff, C. D. Varieties of anger and the inverse link between education and inflammation: toward an integrative framework. Psychosom. Med. 75, 566–574 (2013).

37. Sin, N. L., Graham-Engeland, J. E. & Almeida, D. M. Daily positive events and inflammation: findings from the National Study of Daily Experiences. Brain. Behav. Immun. 43, 130–138 (2015).

38. Fleshner, M. & Crane, C. R. Exosomes, DAMPs and miRNA: Features of Stress Physiology and Immune Homeostasis. Trends Immunol. 38, 768–776 (2017).

39. Magnusson Hanson, L. L. et al. Work stress, anthropometry, lung function, blood pressure, and blood-based biomarkers: a cross-sectional study of 43,593 French men and women. Sci. Rep. 7, 9282 (2017).

Materials & Methods

The authors present many detailed analyses which I appreciate. Next, more detail is needed on the handling on the variables, regarding the way in which the independent variables (i.e. determinants) have been categorised. Simply denoting occupations as “disadvantaged occupations”, “middle occupations”, and “advantaged occupations” is not reproducible and should be substantiated. Regarding education, the authors should definitely use an international classification like the international standard classification of education (ISCED).

Regarding the outcome, CRP: did the laboratories concerned use similar laboratory standards, i.e. how occurred the calibration?

As detailed above in the revised version of the manuscript we’ve provided more details on how occupation related-variables were derived.

This has now been clarified in the life course SEP method section (see p 17) where we provided the following description

“Childhood SEP was ascertained using father’s occupational position reported by the participants and recoded according to the European Socioeconomic Classification (E-SeC) where occupations are classified according to their employment relations and work conditions. We applied 3 E-SeC categories: “disadvantaged occupations” [lower clerical, services, and sales workers, skilled workers,

semi-skilled and unskilled workers, E-SeC class 7-9], “middle occupations” [small employers and self-employed, farmers, lower supervisors and technicians, E-SeC class 4-6], and “advantaged occupations” [higher professionals and managers, higher clerical, services, and sales workers, E-SeC class 1-3].”

and detailed the entire section.

Since educational systems are country-specific and each cohort study collected data in different ways, three comparable levels across countries were identified: “primary or lower secondary school” (from 7 to 9 years after kindergarten with a basic curriculum in languages, mathematics and other subjects); “higher secondary school” (around 4-5 years more, high school diploma level) and “tertiary education” (any degree after high school, such as BSc, MSc, and further education) than can be approximated to an ISCED level from 0 to 2 for the “primary or lower secondary school”; ISCED level from 3 to 5 for the “higher secondary school” and ISCED level greater than 5 for the “tertiary education” category. For example, primary education was not kept separate from low secondary, because this information was available only for few cohorts enrolled in LifePath.

We have acknowledged this in the discussion (paragraph on limitations) together with a sensitivity analyse using binary indicator for occupation related variables (father’s occupational position and participant’s last occupation).

References

Regarding references, some of them are really very old without being classics, e.g. ref. 18 (1997) and ref. 40 (1974). Ref. 19 and 22 seem to misspell the name of Soc Sci Med (adding 1982 to it) and ref. 24 is from 2016 but still in press. Ref. 37 misses information on its publication source.

We thank the reviewer for pointing this out. References are now corrected.

Ref. 18 (1997) has been replaced by

Petrovic, D. et al. The contribution of health behaviors to socioeconomic inequalities in health: A systematic review. Prev. Med. 113, 15–31 (2018).

Name of Soc Sci Med has been corrected and publication source has been added to Vineis, P. et al. reference.

Reviewer #3 (Remarks to the Author):

Overall comments

The authors of this paper show a consistent inverse association between low socioeconomic position (SEP) across life and higher levels of CRP in 4–6 cohorts (3 from the UK, 2 from Switzerland and 1 from Italy) with up to 23,000 participants. There is considerable interest in this field both from scientists interested in how social factors get ‘under the skin’ and those interested in common life course processes underlying ageing. The associations revealed are not novel but this multi-cohort study strengthens the evidence base.

We thank the reviewer for this positive feedback.

There has been a previous systematic review of SEP and CRP in 2007 which the authors only cite in the discussion (Nazmi and Victoria, their ref 28) which looked at over 30 studies (including 4 with early life SEP), reached similar conclusions, and recommended more studies of early SEP and adult inflammation. I would suggest the authors should cite this study in the introduction and explain how their research builds on this systematic review.

In the revised version of the manuscript we refer in greater detail to the systematic review on SEP and CRP published in 2007 and other studies through the following paragraph (p 4)

“A number of studies highlight socioeconomic disadvantage as an upstream determinant of increased basal inflammation. A systematic review of 25 population based studies reported that low SEP mainly assessed by education was associated with elevated CRP level in adulthood across countries⁹. Elevated levels of others circulating inflammatory markers were also reported in disadvantaged socioeconomic groups in general^{10–13} but one reporting gender differences¹⁴. A recent meta-analysis of 15 studies focusing on SEP in childhood revealed an inverse association between early life SEP through parental education or occupation and adulthood CRP¹⁵. ”

9. Nazmi, A. & Victoria, C. G. Socioeconomic and racial/ethnic differentials of C-reactive protein levels: a systematic review of population-based studies. BMC Public Health 7, 212 (2007).

10. Stepanikova, I., Bateman, L. B. & Oates, G. R. Systemic Inflammation in Midlife: Race, Socioeconomic Status, and Perceived Discrimination. Am. J. Prev. Med. 52, S63–S76 (2017).

11. Davillas, A., Benzeval, M. & Kumari, M. Socio-economic inequalities in C-reactive protein and fibrinogen across the adult age span: Findings from Understanding Society. Sci. Rep. 7, 2641 (2017).

12. West, D. A. et al. Life-course origins of social inequalities in adult immune cell markers of inflammation in a developing southern Chinese population: the Guangzhou Biobank Cohort Study. BMC Public Health 12, 269 (2012).

13. Fraga, S. et al. Association of socioeconomic status with inflammatory markers: a two cohort comparison. Prev. Med. 71, 12–19 (2015).

14. Gruenewald, T. L., Cohen, S., Matthews, K. A., Tracy, R. & Seeman, T. E. Association of socioeconomic status with inflammation markers in black and white men and women in the Coronary Artery Risk Development in Young Adults (CARDIA) study. Soc. Sci. Med. 69, 451–459 (2009).

15. Liu, R. S. et al. Socioeconomic status in childhood and C reactive protein in adulthood: a systematic review and meta-analysis. J. Epidemiol. Community Health 71, 817–826 (2017).

We have also added the following sentences at the end of the introduction to better explain how our research builds on the current research context:

“In the available literature on the relationship between SEP and inflammation, the influence played by country and period specific contexts on the social patterning of inflammatory response has been given limited attention due to the lack of available data and/or cross-country variable harmonisation. Investigating the temporal nature of social exposures over the life course and the inflammatory response later in life also needs to be better elucidated.” (Third paragraph of the introduction, p 5).

Their acknowledgement of the previous literature in the discussion is also somewhat limited and undermines the novel aspects that their study brings. For example, I am aware that life course SEP and adult inflammatory markers (both CRP and IL-6) and the role of BMI have been investigated in the 1946 British birth cohort (Jones R, et al. Novel Coronary Heart

Disease Risk Factors at 60-64 years and Life Course Socioeconomic Position. *Atherosclerosis*. 2015 Jan;238(1):70-6) with similar results, yet this paper (and maybe others) are not cited, and I suggest they should be. Similarly, in the discussion, acknowledging that gender differences in lifetime SEP exposure on adult BMI have been previously published would help them explain the gender differences in the associations between SEP and inflammation, and the stronger attenuation by BMI in women.

We thank the reviewer for raising this point. The discussion has been modified accordingly, we've added the following sentence

"Other studies have reported that BMI attenuates the relationships between SEP and CRP^{31,32}."

31. Camelo, L. V. et al. Life course socioeconomic position and C-reactive protein: mediating role of health-risk behaviors and metabolic alterations. The Brazilian Longitudinal Study of Adult Health (ELSA-Brasil). PLoS One 9, e108426 (2014).

32. Jones, R. et al. Novel coronary heart disease risk factors at 60-64 years and life course socioeconomic position: the 1946 British birth cohort. Atherosclerosis 238, 70–76 (2015).

In regards to the statistical analysis, if, as the authors indicate, they are taking a life course approach to investigate the association between lifetime SEP and adult inflammation, could the authors explain why they do not use the structured life course modelling approach (which has been used by several groups) to select the life course models that best fit the data, rather than a simple set of regression models? Also gender differences are reported and commented upon but no tests of interaction are mentioned. A statistical reviewer may have other points but this is not my area of expertise.

We are using the life course approach, which is a conceptual framework grounded in an interdisciplinary literature from the social sciences and epidemiology [Kelly-Irving M, Tophoven S, Blane D. Life course research: new opportunities for establishing social and biological plausibility. Int J Public Health. 2015 Sep;60(6):629-30. doi: 10.1007/s00038-015-0688-5. PubMed PMID: 25981211.]. Our approach consists of taking into account the chronology of exposures over the life course, to understand how they are associated with inflammation. The life course framework facilitates the interpretation of these statistical analyses.

The following sentence has been added in the last paragraph of the introduction (p. 5) to address the reviewers point.

"Our approach consists of taking into account the chronology of exposures over the life course, to understand how they are associated with inflammation."

The results, figures and tables could do with clarification in places and better labelling (see below).

We have improved the quality of the graphs – mainly Figure 2 and Supplementary Figures 1-3. Table 2 and 3 have also been clarified in the revised version of the manuscript.

Table 2 in the revised version of the manuscript has been transposed to give in columns each SEP indicators and in rows corresponding results for each model. A detailed footnote has also been added to better describe the table

[†]Except for model 1 + sedentary where N = 7 769

[‡]Except for model 1 + sedentary where N = 17 699

^{*}Model 1 adjusted for age and sex

[#]Model 2 controlled for age, sex and additionally alcohol, smoking, BMI and sedentary

Abbreviations: CI, confidence interval; I^2 , percentage of between study heterogeneity; P_H , P-value of heterogeneity test; BMI, body mass index.“

A detailed footnote has been added to Table 3 to better describe the linear regression models

*^{**}Model A adjusted for age, sex, father’s occupational position*

^{††}Model B adjusted for age, sex, father’s occupational position and participant’s educational attainment

^{‡‡}Model C adjusted for age, sex, father’s occupational position and participant’s last occupation

^{§§}Model D adjusted for age, sex, father’s occupational position, participant’s educational attainment and participant’s last occupation

^{¶¶}Fully adjusted model controlled age, sex, father’s occupational position, participant’s educational attainment, participant’s last occupation and additionally alcohol, smoking, BMI and sedentary

Abbreviations: CI, confidence interval; I^2 , heterogeneity; P_H , P-value of heterogeneity test; BMI, body mass index.“

The discussion is wordy and could be tightened considerably, and repetitions removed.

In the revised version of the manuscript, most of the discussion has been rewritten to avoid repetitions, to be more informative while staying concise.

Previous studies that support (or not) the findings of this study should be adequately cited.

We thank the reviewer for the helpful suggestion and have rewritten the discussion accordingly, more specifically in the second paragraph.

So overall this is a useful study because of the multi-cohort features but it somewhat undermines itself by not showing how it builds on previous literature. It confirms previous evidence rather than influencing new thinking in this field.

We thank the reviewer for this comment.

The conclusions need to be strengthened to be more precise about the overall strength of the evidence and the further research needed and then what the clinical or policy implications could/should be.

In the revised version of the manuscript, we have added the following sentence at the very end of the discussion to strengthen our message

“In conclusion, our study highlights the important role that, from early life, social factors play in health beyond behaviours and lifestyle factors. We document the biological consequences – through inflammation – of social disadvantages, justifying the need to intervene from early life to reduce social disparities in health. Further work integrating biosocial perspectives in longitudinal settings is needed to better understand the mechanisms through which the social environment shapes physiological processes that are important for complex health outcomes.”

Specific points

Abstract: The abstract would be strengthened by: including a brief rationale for studying the association between lifetime SEP and later CRP; including the key estimates from the

regression models; making the last sentence less vague. (see also typo 'adpoting')

In the revised manuscript, as suggested by Reviewer 3, we have strengthened the abstract by including a brief rationale

"Chronic inflammation has been proposed to have a prominent role in the construction of social inequalities in health. Disentangling the effects of early life and adulthood social disadvantage on inflammation is key in elucidating biological mechanisms underlying socioeconomic disparities."

In addition, we have revised the last sentence to make it less vague:

"These findings suggest socioeconomic disadvantage in young adulthood is independently associated with later life inflammation calling for further studies of the pathways operating through educational processes."

We thank you for catching the typos. To provide a non-technical summary of main findings we prefer rather not include key estimates from the regression model.

Introduction:

p.4-5, lines 92-93: Suggest the phrase ' the social patterning of the life course response remains poorly described in terms of life course stage' needs toning down (given overall comments above) and relevant references should be cited here and then commented upon in the discussion.

We've re-written this sentence as

"Investigating the temporal nature of social exposures over the life course and the inflammatory response later in life also needs to be better elucidated."

in the revised version of the manuscript (p 5).

Last paragraph: the introduction would be strengthened by: stating some hypotheses based on what we already know about this topic; and by making clear that when the analyses were performed separately for men and women, tests of sex differences would also be conducted.

The last paragraph of the introduction has been reformulated in the revised version of the manuscript to address the reviewer's concerns (p 5).

Results

First para of text. Figure 1 needs improving. What the mean age refers to should be clear in the figure – it appears from the text to be age at enrolment but that is not correct for NCDS at least. Include the standard deviation as well as the mean CRP level.

This has now been clarified. An explicit footnote has been added to Figure 1 legend

*“*NCDS is the only birth cohort, therefore father’s occupation was collected prospectively and mean age corresponds to the age of participants at the time of the biomedical survey.”*

And both CRP mean and standard deviation has been added to the Figure 1.

How do those included in the samples (ranging from 6%-70%) differ from those who were not included in the samples?

We have added in the Supplementary Information (Supplementary Table 1) a table with characteristics and life course socioeconomic position for the full population and for the study population. We have also added a statement in the Study population paragraph:

“Key characteristics of each study population showed small to moderate differences compared to our analytical sample (Supplementary Table 1).”

Perhaps also make clear which measures of father’s occupation were prospective and retrospective.

This has now been clarified. Except in the NCDS birth cohort, measure of father’s occupation were retrospective.

ELSA CRP levels are almost twice as high compared with others – is this the result of their higher mean age?

We think so. It has been shown that levels of CRP increase with age. [Woloshin S, Schwartz LM. Distribution of C-reactive protein values in the United States. N Engl J Med. 2005 Apr 14;352(15):1611-3. PubMed PMID: 15829550.] ELSA participants included in our study are the oldest and in average 67.4 years old; from 15 to 20 years older than participants from the other cohorts.

Line 128 subheading ‘ Social effect of life course SEP on CRP’ is a rather curious heading. And it is not clear how it differs from the second subheading ‘life course SEP and CRP’. Rephrase?

We agree with the Reviewer 3 that our results subsections headings were rather confusing and have corrected them accordingly “Association between each respective SEP indicator and CRP” and “Contribution of SEP at each stage of the life course to adult CRP”

Line 129-160 I found these three paragraphs hard to follow with the relevant tables. The title of the tables need better labelling (so it is clear they are showing the various adjusted models) and better signposting in the text.

Table 2 and 3 have also been clarified in the revised version of the manuscript and the result section has been re-written in the revised version of the manuscript, more specifically we have added subheads to ease the understanding.

When the text indicates that the associations are stronger in men than women (or vice versa) have tests of interaction been performed – perhaps give relevant statistic in brackets?

Although none of the interaction tests performed between each SEP and gender within each cohort was found nominally significant in our multivariate models (data not shown), evidence is accumulating on gender differences in health behaviours, in stress responses and also in diseases onset and/or progression. We added a justification to report our results stratified by gender in the Method section (p 19-20) and we are no longer using the following type of formulations “associations were stronger in men than women”.

Discussion

216-217 “to mimic life course experiences’ needs clarifying given that the materials and methods come later.

This has now been corrected. We’ve replaced ‘to mimic life course experiences’ by ‘whereby life course SEP variables were added to the model in chronological order’

Line 230-231 – reference to previous literature not sufficient (see overall comments). Whole paragraph is wordy, needs tightening.

We thank the reviewer for pointing this out, we have extended our paragraph aiming to compare our work with previous studies as follow:

“Overall, our results are consistent with previous studies investigating the relationship between SEP at different life stages and chronic inflammation. A previous systematic review of published observational studies up to 2006 reported associations between adult SEP and CRP ⁹. Since then the same relationship has also been demonstrated in various other observational studies from worldwide countries (Taiwan ²⁷, Europe ¹³, Brazil ³¹ and others ^{17,28–30}). A recent meta-analysis of population-based and cross-sectional studies showed that low childhood SEP was associated with higher adulthood CRP ¹⁵.” (Second paragraph of the discussion in the revised version of the manuscript, p 11)

9. Nazmi, A. & Victora, C. G. Socioeconomic and racial/ethnic differentials of C-reactive protein levels: a systematic review of population-based studies. *BMC Public Health* 7, 212 (2007).

27. Lin, Y.-H., Jen, M.-H. & Chien, K.-L. Association between life-course socioeconomic position and inflammatory biomarkers in older age: a nationally representative cohort study in Taiwan. *BMC Geriatr.* 17, 201 (2017).

13. Fraga, S. et al. Association of socioeconomic status with inflammatory markers: a two cohort comparison. *Prev. Med.* 71, 12–19 (2015).

31. Camelo, L. V. et al. Life course socioeconomic position and C-reactive protein: mediating role of health-risk behaviors and metabolic alterations. *The Brazilian Longitudinal Study of Adult Health (ELSA-Brasil). PLoS One* 9, e108426 (2014).

17. Pollitt, R. A. et al. Early-life and adult socioeconomic status and inflammatory risk markers in adulthood. *Eur. J. Epidemiol.* 22, 55–66 (2007).

28. Koster, A. et al. Association of inflammatory markers with socioeconomic status. *J. Gerontol. A. Biol. Sci. Med. Sci.* 61, 284–290 (2006).

29. Loucks, E. B. et al. Life course socioeconomic position is associated with inflammatory markers: the Framingham Offspring Study. *Soc. Sci. Med.* 71, 187–195 (2010).

30. Stringhini, S. et al. Association of lifecourse socioeconomic status with chronic inflammation and type 2 diabetes risk: the Whitehall II prospective cohort study. *PLoS Med.* 10, e1001479 (2013).

Line 245-258 – finding that BMI attenuates the associations most is not a novel finding and other studies should be cited to provide the strength of the evidence overall.

Line 289-291 There are a number of papers that need citing that show that the accumulation of SEP exposures across life on adult overweight/obesity are stronger for women than men which may explain the gender differences and also why the attenuation by BMI was stronger in women than men.

The discussion has been modified accordingly, we've added the following sentence

"Other studies have reported that BMI attenuates the relationships between SEP and CRP^{31,32}.

31. Camelo, L. V. et al. Life course socioeconomic position and C-reactive protein: mediating role of health-risk behaviors and metabolic alterations. The Brazilian Longitudinal Study of Adult Health (ELSA-Brasil). PloS One 9, e108426 (2014).

32. Jones, R. et al. Novel coronary heart disease risk factors at 60-64 years and life course socioeconomic position: the 1946 British birth cohort. Atherosclerosis 238, 70–76 (2015).

Line 303-307: so how did those with missing data in the cohorts differ from those with complete covariate data?

An additional table with characteristics and life course socioeconomic position for the full population and for the study population has been added in the revised manuscript (Supplementary Table 1). We have also added a statement in the Study population paragraph:

"Key characteristics of each study population showed small to moderate differences compared to our analytical sample (Supplementary Table 1)."

L319-321 A definite limitation is the use of CRP as the only inflammatory marker. This limitation needs more discussion (e.g. in relation to the inflammatory cascade) and whether other inflammatory markers (like IL-6) could be more informative.

We agree with the reviewer and the following sentences has been added in the discussion:

" While CRP is a useful proxy for overall inflammatory status, we would ideally have used it alongside Tumour Necrosis Factor alpha and Interleukin 6 for example, in order to better capture the complex regulatory cascades involved in inflammation. The inflammatory response is coordinated by a number of cytokines as well as signalling proteins of the immune system⁴⁷. Future investigations on inflammatory related biomarkers at wider and larger scale are needed to disentangle the complex inflammatory responses associated to pathological and immunological processes"

47. Ashley, N. T., Weil, Z. M. & Nelson, R. J. Inflammation: Mechanisms, Costs, and Natural Variation. Annu. Rev. Ecol. Evol. Syst. 43, 385–406 (2012).

Line 325-329: so were the SEP indicators associated with BMI in each of the cohorts?

The Supplementary Table 2 gives a description of each cohort by life course SEP. For the available life course SEP in each cohort, we observe a (unadjusted) significant association between SEP and BMI

categorised in 4 classes where participants with a disadvantaged SEP were more likely to be overweight and/or obese.

Line 330-337: the last paragraph of the discussion raises rather vague needs for further research. Can the authors be more precise and also comment on any clinical or policy implications. Some overall assessment of the strength of the evidence (not just their study) would be helpful.

As mentioned above, we have added the following sentences at the very end of the discussion:

“In conclusion, our study highlights the important role that, from early life, social factors play in health beyond behaviours and lifestyle factors. We document the biological consequences – through inflammation – of social disadvantages, justifying the need to intervene from early life to reduce social disparities in health. Further work integrating biosocial perspectives in longitudinal settings is needed to better understand the mechanisms through which the social environment shapes physiological processes that are important for complex health outcomes.”

Materials and methods

359-360: Information about sampling methods and laboratory analyses are not given in Supplementary table 8.

Thanks for pointing this out. Since we’ve updated the Supplementary Information file, the table providing additional information about the Lifepath cohort studies included in our work is now Supplementary Table 15 and this has been corrected in the main revised manuscript.

377-379: the authors say that the choice of intermediate variables was based on previous literature but fail to cite these studies.

We have added several references in the revised version of the manuscript in the Methods sections to better strengthen the justification of the intermediate factors used in our statistical analyses.

“The following factors have been found to be associated with both SEP and CRP in the literature and were therefore considered as intermediate variables that may mediate the relationship between SEP and CRP: BMI^{55,56}[...]; smoking status^{34,57}[...]; alcohol consumption^{58,59}[...]; sedentary^{60,61}[...].”

34. Shiels, M. S. et al. Cigarette Smoking and Variations in Systemic Immune and Inflammation Markers. *JNCI J. Natl. Cancer Inst.* 106, dju294–dju294 (2014).

55. McLaren, L. Socioeconomic status and obesity. *Epidemiol. Rev.* 29, 29–48 (2007).

56. Choi, J., Joseph, L. & Pilote, L. Obesity and C-reactive protein in various populations: a systematic review and meta-analysis. *Obes. Rev. Off. J. Int. Assoc. Study Obes.* 14, 232–244 (2013).

57. Giskes, K. et al. Trends in smoking behaviour between 1985 and 2000 in nine European countries by education. *J. Epidemiol. Community Health* 59, 395–401 (2005).

58. González-Reimers, E., Santolaria-Fernández, F., Martín-González, M. C., Fernández-Rodríguez, C. M. & Quintero-Platt, G. Alcoholism: a systemic proinflammatory condition. *World J. Gastroenterol.* 20, 14660–14671 (2014).

59. Mackenbach, J. P. et al. Inequalities in Alcohol-Related Mortality in 17 European Countries: A Retrospective Analysis of Mortality Registers. *PLoS Med.* 12, e1001909 (2015).

60. Hamer, M. et al. *Physical activity and inflammatory markers over 10 years: follow-up in men and women from the Whitehall II cohort study. Circulation* 126, 928–933 (2012).

61. O'Donoghue, G. et al. *Socio-economic determinants of physical activity across the life course: A 'DEterminants of Diet and Physical ACTivity' (DEDIPAC) umbrella literature review. PloS One* 13, e0190737 (2018).

Statistical analysis

Nowhere does it indicate that sex interaction tests were carried out. Please clarify as statements that some associations were stronger in men than women (or vice versa) are made in the text.

We thank the reviewer for pointing this out. Interaction tests were carried out but not significant (data not shown), we justified in the Method section in the revised version of the manuscript our choice to present the results also stratified by gender. It has also been clarified in the Results and Discussion section. We also carefully checked to avoid writing 'associations were stronger in men than women (or vice versa)'..

Line 402-416: Please explain whether the structured life course modelling approach would have been preferable to models A-C to test whether lifetime SEP exposures were cumulative or linked to sensitive periods.

As mentioned above, we took a life course approach to guide our chronological analysis of the exposures and interpretations. While some authors have attempted to examine specific hypothesised mechanisms such as sensitive periods and accumulation, we chose to take into account the timing of social exposures by examining them separately and when adjusted sequentially. We did not aim hypothesise about sensitive periods and accumulation, when each social variable represents a relatively wide period of the life course, and is related to the subsequent social exposures.

Reviewers' Comments:

Reviewer #1:

Remarks to the Author:

The authors have addressed all of my comments.

Reviewer #2:

Remarks to the Author:

Nature Communications manuscript NCOMMS-18-20248A

The authors in general did a good job in handling my comments. Some remaining issues still have to be solved, I list them just in the sequence of the rebuttal. My reviewing of the rebuttal regards both my own comments (Rev#2) and those of Rev#3 as requested by the Editor. My most major comment regards the issue of modification of the associations by gender as noted by Rev#3. These should be handled differently, in my opinion, see my further comment at that point.

R1 The clarification of the categories is sufficient indeed, but having read them my feeling is that the label 'disadvantaged occupations' for ESeC 7-9 is too explicit. This e.g. regards lower grade white collar workers, so I would use a different label.

Next, I also think that a discussion of the rather skewed socioeconomic distributions should be added to the text.

R3, final part; the sentence: '\. . . infectious diseases and conditions (such as Herpes and Epstein Barr infections, oral health),' should be rewritten somewhat – oral health is not an infection or condition, so please reword.

Discussion

The authors moved the discussion on gender differences to another place in the sequence. That paragraph has an interesting story, but the degree to which this builds upon the literature remains really unclear – the full paragraph just contains one reference – more adequate referencing is required here. Moreover, Rev#3 is in my opinion right in noting that this discussion should be reversed given the non-significance of the interactions with gender.

Reviewer 3

The authors provide a new paragraph on previous studies

"A number of studies highlight socioeconomic disadvantage as an upstream determinant of increased basal inflammation. A systematic review of 25 population based studies reported that low SEP mainly assessed by education was associated with elevated CRP level in adulthood across countries 9. Elevated levels of others circulating inflammatory markers were also reported in disadvantaged socioeconomic groups in general 10–13 but one reporting gender differences 14. A recent meta-analysis of 15 studies focusing on SEP in childhood revealed an inverse association between early life SEP through parental education or occupation and adulthood CRP 15. "

I am missing a message here, in particular on what this study adds to what we know based on these previous studies. This may be partially due to the fact that the gaps in knowledge have been listed in the introduction. Why not combine this paragraph from the Discussion with that, and move it to the Introduction?

Similarly, I am missing the 'so what' for the added sentence "Other studies have reported that BMI attenuates the relationships between SEP and CRP 31,32."

The issue of assessing gender differences as an interaction term in the statistical modelling should be solved. The authors indicate at the end of their rebuttal that these interactions are not statistically significant. In that case, that should be mentioned, and all interpretation of these differences should be left out – they do not exist simply, so the discussion should focus on why they do not occur. So I would suggest:

- start the Results part on Gender effect (line 181 and further) with an adequate presentation of the (negative) findings regarding the non-significance of all interaction terms with gender, and show the significance levels of these interaction terms
- omit any result that has not been substantiated as differing by gender with statistical significance
- fully revise the discussion on gender effects from line 284 onwards (till 292) – that an effect size is larger for men than women is most likely chance if having such huge pooled samples
- similarly revise the Methods section on this gender effect (lines 453-459).

Small textual issues:

The new parts of the Results sometimes use present tense, the original parts past tense
The sentence 'but one reporting gender differences' (line 14) misses some words.

Response to the reviewers' comments on manuscript NCOMMS-18-20248A

Reviewer 2

The authors in general did a good job in handling my comments. Some remaining issues still have to be solved, I list them just in the sequence of the rebuttal. My reviewing of the rebuttal regards both my own comments (Rev#2) and those of Rev#3 as requested by the Editor. My most major comment regards the issue of modification of the associations by gender as noted by Rev#3. These should be handled differently, in my opinion, see my further comment at that point.

* The clarification of the categories is sufficient indeed, but having read them my feeling is that the label 'disadvantaged occupations' for ESeC 7-9 is too explicit. This e.g. regards lower grade white collar workers, so I would use a different label.

-Response: We suggest changing the formulation to "less advantaged occupations"; "middle occupations"; "more advantaged occupations" and have modified this accordingly throughout.

* Next, I also think that a discussion of the rather skewed socioeconomic distributions should be added to the text.

Response: In addition to having added "in particular because of a potential under-representation of the most advantaged/ less disadvantaged SEP categories" previously, we also addressed this point by adding this to pg 15: "Additionally, the distribution of the SEP measures was skewed"

*R3, final part; the sentence: '.. infectious diseases and conditions (such as Herpes and Epstein Barr infections, oral health),' should be rewritten somewhat – oral health is not an infection or condition, so please reword.

Response: we addressed this by reformulating the sentence pg 11:

"These include, for example, chemical exposures, infectious diseases (such as Herpes and Epstein Barr infections etc) and oral health conditions, as well as psychosocial stress."

*Discussion: The authors moved the discussion on gender differences to another place in the sequence. That paragraph has an interesting story, but the degree to which this builds upon the literature remains really unclear – the full paragraph just contains one reference – more adequate referencing is required here. Moreover, Rev#3 is in my opinion right in noting that this discussion should be reversed given the non-significance of the interactions with gender.

Response: We have added references to this section on pg 13, and altered a sentence:

"Overall, we found similar patterns in the relationship between SEP and CRP for men and women, though the effect size between father's occupation and educational attainment and CRP were higher in women. Some differences between men and women may come down to differences in immunological and inflammatory responses influenced by both sex and gender. Sex contributes to

differences that influence the deposit of adipose tissue, its physiological interaction with the endocrine system, and associated metabolic risk over the lifecourse⁴³. Gender may reflect behaviours that influence exposure to microorganisms, access to healthcare or health-related behaviours that affect health trajectories⁴⁴. We also observed that the attenuation of the relationship once BMI was introduced into the models appeared to be greater among women. This may suggest that inflammation induced through greater body mass and adipose tissue was a more significant pathway for women from disadvantaged SEP groups⁴⁵.”

Reviewer 3

*The authors provide a new paragraph on previous studies

“A number of studies highlight socioeconomic disadvantage as an upstream determinant of increased basal inflammation. A systematic review of 25 population based studies reported that low SEP mainly assessed by education was associated with elevated CRP level in adulthood across countries 9. Elevated levels of other circulating inflammatory markers were also reported in disadvantaged socioeconomic groups in general 10–13 but one reporting gender differences 14. A recent meta-analysis of 15 studies focusing on SEP in childhood revealed an inverse association between early life SEP through parental education or occupation and adulthood CRP 15. ”

I am missing a message here, in particular on what this study adds to what we know based on these previous studies. This may be partially due to the fact that the gaps in knowledge have been listed in the introduction. Why not combine this paragraph from the Discussion with that, and move it to the Introduction?

Response: This paragraph mentioned by the reviewer was already placed in the introduction, to better explain how our research builds on the current research context. We refer to two meta-analyses in the discussion as well to illustrate how our results fit within the findings from previous studies.

*Similarly, I am missing the ‘so what’ for the added sentence “Other studies have reported that BMI attenuates the relationships between SEP and CRP 31,32.”

Response: We have added this sentence to pg 12:

“Together these sources of evidence may point towards the accumulation of body fat among more socially disadvantaged populations as being a mechanism leading to higher levels inflammation.”

*The issue of assessing gender differences as an interaction term in the statistical modelling should be solved. The authors indicate at the end of their rebuttal that these interactions are not statistically significant. In that case, that should be mentioned, and all interpretation of these differences should be left out – they do not exist simply, so the discussion should focus on why they do not occur. So I would suggest:

-start the Results part on Gender effect (line 181 and further) with an adequate presentation of the (negative) findings regarding the non-significance of all interaction terms with gender, and show the significance levels of these interaction terms

Response: we have now added Supplementary Table 16 reporting interaction results in the Methods section

-omit any result that has not been substantiated as differing by gender with statistical significance

Response: Given the potential social and biological differences between men and women we feel it is justified to report of the stratified analyses by gender, having shown no interaction by gender.

-fully revise the discussion on gender effects from line 284 onwards (till 292) – that an effect size is larger for men than women is most likely chance if having such huge pooled samples

Response: This has been checked and amended when necessary.

-similarly revise the Methods section on this gender effect (lines 453-459).

Response: We have added a reference to the interaction analyses and reference to Supplementary Table 16.

*Small textual issues:

-The new parts of the Results sometimes use present tense, the original parts past tense

Response: This has been addressed

The sentence 'but one reporting gender differences' (line 14) misses some words.

Response: Thank you, we have changed this to "Elevated levels of others circulating inflammatory markers were also reported in disadvantaged socioeconomic groups in general¹⁰⁻¹³ and also regarding gender differences¹⁴."